# Enhancer accessibility and CTCF occupancy underlie asymmetric TAD architecture and cell type specific genome topology

Christopher Barrington[1,3], Dimitra Georgopoulou[1,4], Dubravka Pezic[1], Wazeer Varsally[1], Javier Herrero [2] & Suzana Hadjur[1]

Cohesin and CTCF are master regulators of genome topology. How these ubiquitous proteins contribute to cell-type specific genome structure is poorly understood. Here, we explore quantitative aspects of topologically associated domains (TAD) between pluripotent embryonic stem cells (ESC) and lineage-committed cells. ESCs exhibit permissive topological configurations which manifest themselves as increased inter- TAD interactions, weaker intra-TAD interactions, and a unique intra-TAD connectivity whereby one border makes pervasive interactions throughout the domain. Such 'stripe' domains are associated with both poised and active chromatin landscapes and transcription is not a key determinant of their structure. By tracking the developmental dynamics of stripe domains, we show that stripe formation is linked to the functional state of the cell through cohesin loading at lineage-specific enhancers and developmental control of CTCF binding site occupancy. We propose that the unique topological configuration of stripe domains represents a permissive landscape facilitating both productive and opportunistic gene regulation and is important for cellular identity.

---

[1] Research Department of Cancer Biology, University College London, Paul O'Gorman Building, 72 Huntley Street, London WC1E 6BT, UK. [2] Bill Lyons Informatics Centre, University College London, Paul O'Gorman Building, 72 Huntley Street, London WC1E 6BT, UK. [3] Present address: Bioinformatics, The Francis Crick Institute, 1 Midland Road, London NW1 1AT, UK. [4] Present address: CRUK Cambridge Institute, University of Cambridge, Li Ka Shing Centre, Robinson Way, Cambridge CB2 0RE, UK. Correspondence and requests for materials should be addressed to S.H. (email: s.hadjur@ucl.ac.uk)

Mammalian genomes are folded into structures known as topologically associating domains (TADs) that facilitate the regulation of gene expression[1]. This is achieved through the spatial clustering of regulatory elements with appropriate target genes and by creating permissive topologies within which lineage-appropriate regulators can emerge and function[2]. Hi-C studies across many cell types have shown that TADs are similarly positioned in pluripotent and differentiated cells[3,4], notwithstanding the dramatic epigenome reorganisation and transcriptional changes which accompany these transitions in cellular potential. Building upon these observations, recent studies have uncovered quantitative differences in genome topology during development and reprogramming[5–7]. Changes in the extent of TAD insulation, as well as TAD connectivity have been associated with gene expression changes accompanying developmental[8–10] or disease progressions[11–13].

Whilst other factors can play important roles in spatial organisation[14–16], it is well established that CTCF and cohesin are master regulators of TAD structure. These proteins are key components of genome topology at multiple scales, and thus have central roles in the connectivity and insulation of TADs[17–21]. How these ubiquitously expressed structural proteins mediate genome topologies that reflect cellular potential is poorly understood. In line with their importance in spatial genome organisation, polymer models of TAD formation incorporating cohesin-mediated loop extrusion and CTCF boundary elements gives rise to TAD structures that are very similar to experimental Hi-C datasets[22,23]. While these models do consider the occupancy of a CTCF-binding site as a parameter, it remains unclear how ChIP signals relate to binding site occupancy and how differentially occupied binding sites contribute to cell-type-specific topologies.

To explore how genome topology relates to cellular potential, here we explore quantitative aspects of TAD structure between pluripotent embryonic stem cells (ESC) and lineage-committed cells. Our results reveal chromatin interactions in ESCs which become increasingly restricted with lineage commitment, in line with the known chromatin plasticity of ESCs. This permissivity manifests itself as increased interactions between TADs, weaker interactions within TADs and a class of TAD architecture with a unique connectivity whereby an individual element at one border makes pervasive interactions throughout the domain. Interestingly, such TAD structure is not unique to ESCs and is also readily observed in lineage-restricted cell types. A similar TAD architecture, so-called stripe domains has recently been described[24]. Here we track the developmental dynamics of stripe domains to show that apart from being merely structural constructs, stripe domain formation is linked to the functional state of the cell through cohesin loading at lineage-appropriate enhancer elements and developmental control of CTCF-binding site occupancy. Importantly, our results indicate that transcription is not a key determinant of stripe conformation, since stripe domains can be associated with poised enhancers and low levels of gene expression. We find that enhancer activation during lineage commitment contributes to stripe conformation and that lineage-dependant regulation of CTCF-binding site occupancy converts stripe domains into loop domains. Overall our results indicate that cell-type-specific control of genome topology may be achieved by coupling cohesin loading to lineage-appropriate enhancers and by regulating the occupancy of CTCF at specific genomic locations. Since both active and poised enhancers are associated with stripe domains, we propose that the unique topological configuration of these TADs make them well suited to a permissive structural environment within which lineage-appropriate gene regulation may readily emerge.

## Results

### Analysis of genome topology changes during lineage commitment.
To investigate quantitative changes to genome topology in cells of different fates, we prepared Hi-C libraries as previously reported[17,25] from biological replicates of mouse ESC grown in 2i conditions and compared these to mouse neural stem cell (NSC) Hi-C datasets[6,17]. Sequencing of the ESC libraries produced 1.6 billion paired-end reads, with 319 million total contact pairs retained after filtering (Supplementary Fig. 1a–f, Supplementary Table 1). We used a bin-free, non-parametric approach (termed Shaman[26]) for normalisation of the Hi-C datasets and calculation of an interaction *score* for each contact in the observed contact matrix (see the section "Methods" for further information). Analysis of contact insulation supported previous findings[4,6] that many TAD borders were similarly positioned between pluripotent and lineage-committed cells (69.4%, when a separation of 50 kb or less is considered) (Supplementary Fig. 1g, h and "Methods"). Further examination revealed quantitative differences to the TAD landscape that accompanied loss of pluripotency and acquisition of the NSC state. We observed increased inter-TAD interactions in ESCs compared to NSCs (Fig. 1a, b) and a corresponding change in the extent of insulation whereby ESCs exhibited weaker TAD border insulation compared to NSCs (Fig. 1c, d, Supplementary Fig. 1i). Moreover, intra-TAD interactions (Fig. 1e, f), TAD connectivity[5] (Fig. 1g) and interactions between convergent CTCF sites (Supplementary Fig. 1j) were significantly enriched in NSC compared to ESCs (KS test, $p < 0.001$). These differences were not a function of a difference in the size of TADs, nor a difference in the overall distribution of interaction scores between the two cell types (Supplementary Fig. 1k, l) and similar quantitative changes to intra-TAD and inter-TAD interactions can be observed in other Hi-C datasets from cells of distinct identity (Supplementary Fig. 1m–q). Furthermore, our observations are supported by published reports of chromatin topology restriction during lineage commitment[6] (Supplementary Fig. 1r–u). Thus, despite similar TAD organisation between the cell types, contact frequencies between and inside TADs are different as cells change fate and thus may play an important role in defining cellular potential.

In addition to these quantitative topological differences, we observed numerous examples of TADs whose borders were maintained between the cell types but which had dramatically different internal TAD connectivity. In contrast to well-characterised loop domain TADs[27] which have specific interactions between their borders (Fig. 1h), we also observed TADs characterised by an asymmetric enrichment of high-scoring interactions anchored from one boundary and interacting with fragments throughout the TAD, producing lines of contact enrichment (Fig. 1i). We reasoned that distinct mechanisms could underlie the differences in connectivity and that this may lead to a new understanding of lineage commitment at the level of genome topology. While preparing our manuscript, a very similar TAD architecture was described, so-called stripe domains[24], thus, we have adopted this nomenclature herein. We explore the relationship between the stripe TADs we annotate and those annotated in prior studies in the section "Discussion".

### Unbiased identification of intra-TAD structures.
To explore the mechanisms that underlie the unique intra-TAD connectivity of stripe domains, we developed a computational approach to comprehensively characterise TAD structures from Hi-C data in an unbiased manner. Briefly, high-scoring interactions (>40) within each TAD (with a minimum size of 200 kb) were grouped into one of four defined TAD sectors corresponding to whether they were observed between TAD borders (border interaction), in

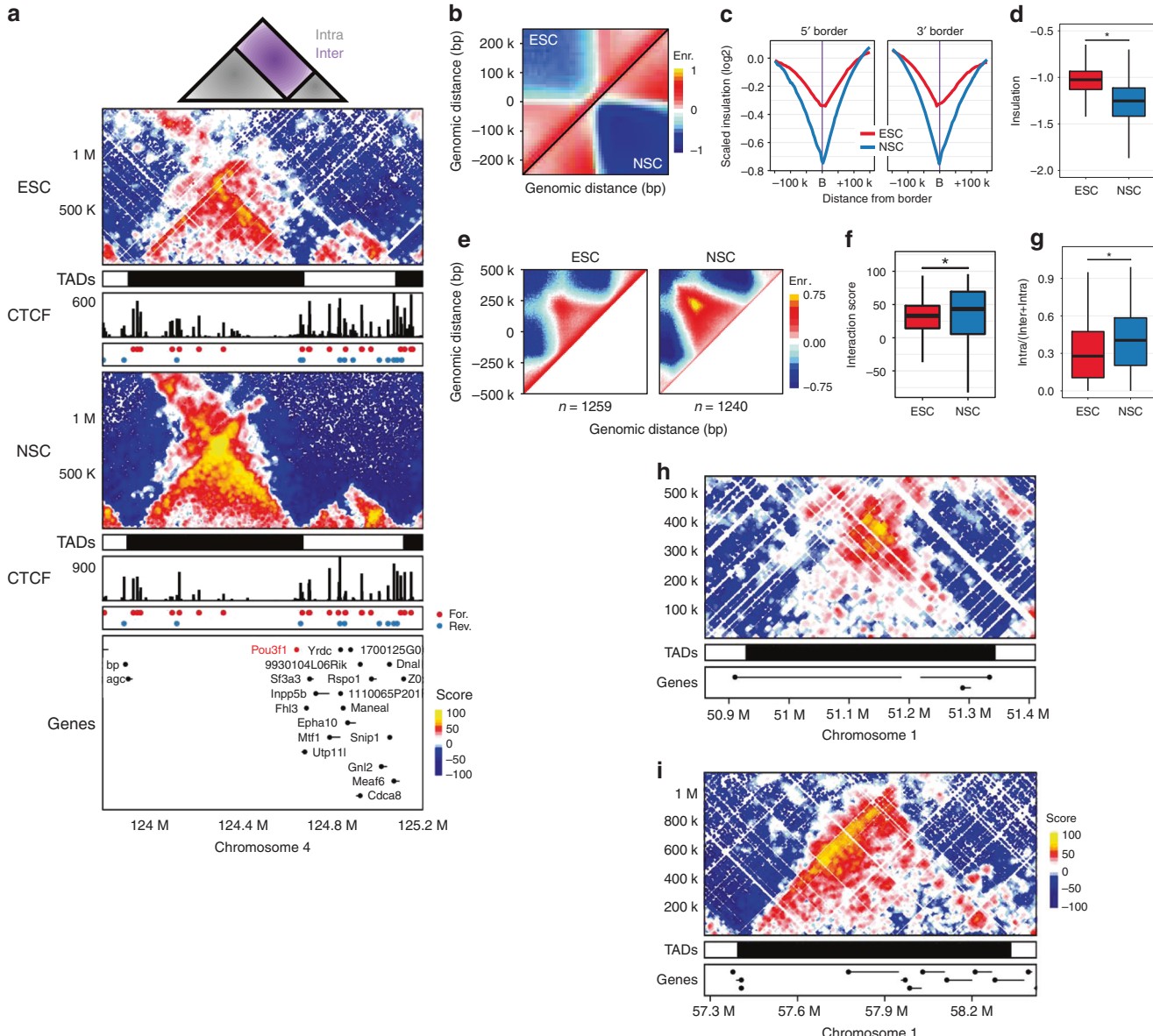

**Fig. 1** Topological changes during lineage commitment. **a** Schematic of intra-TAD (grey) and inter-TAD (purple) interactions (top). Hi-C and CTCF ChIP-seq data for a 1.5 Mb region on chr4 around the *Pou3f1* gene in ESC (upper) and NSC (middle). TADs are represented as alternating black/white rectangles and gene annotations are shown in the bottom panel. Hi-C contact maps show the interaction 'Scores' for individual fragment end pairs, colour-coded according to the density of the observed contacts around it and normalised by the density of the expected contacts (see the section "Methods"). CTCF ChIP-seq tracks as well as colour-coded CTCF motifs under ChIP peaks are shown for both cell types (red and blue dots represent forward and reverse motifs, respectively). **b** Aggregate Hi-C maps of ESC (upper half) and NSC (lower half) TAD borders reveal increased insulation between NSC TADs. **c** Scaled contact insulation profiles across ESC (red) and NSC (blue) TAD borders at 300 kb band. **d** Distribution of observed insulation at borders in ESC (red) and NSC (blue). Central bar represents the median with boxes indicating the upper and lower quartiles. KS test *$p < 0.001$. **e** Aggregate Hi-C maps of size-selected (30th–70th percentiles) TADs from both cell types showing increased interactions between TAD borders in NSC (left panel), note strong 'corner' interaction in NSCs. **f** Distribution of mean interaction score between 20 kb regions centred on intra-TAD border pairs in ESC and NSC (right panel). Box plot as in **d**. **g** Distribution of TAD connectivity defined as the number of high-score contacts (>40) that are located within a TAD as a proportion of high-score contacts that connect the TAD to a 10 Mb region up or downstream of the TAD. Box plot as in **d**. **h, i** Examples of TAD types observed in ESC Hi-C data; **h** TAD with a prominent interaction between the TAD borders (loop domain) and (**i**) TAD with an asymmetric contact profile 'anchored' on the 5′ border (5′ stripe domain). Shown also are the TAD positions and gene annotations as in (**a**)

the 'leading' or 'trailing' edge of the TAD, or in the inner-TAD region (Fig. 2a and "Methods"). Spatial enrichment of high-scoring interactions in a TAD sector was calculated using the *Z* statistic to compare the number of observed high-scoring interactions in a given sector to the number expected by randomly sampling all intra-TAD interactions, irrespective of the interaction score. Enrichment in the TAD sectors was used to classify the

TADs, applying a stringent threshold for classification (Fig. 2a and "Methods").

Aggregate Hi-C maps for each classified TAD group revealed general topology trends for each TAD class (Fig. 2b, Supplementary Fig. 2a). The complete set of ESC TADs that were considered for classification (*n* = 3142) showed the distinctive TAD topology of a self-interacting region flanked by contact insulation defining

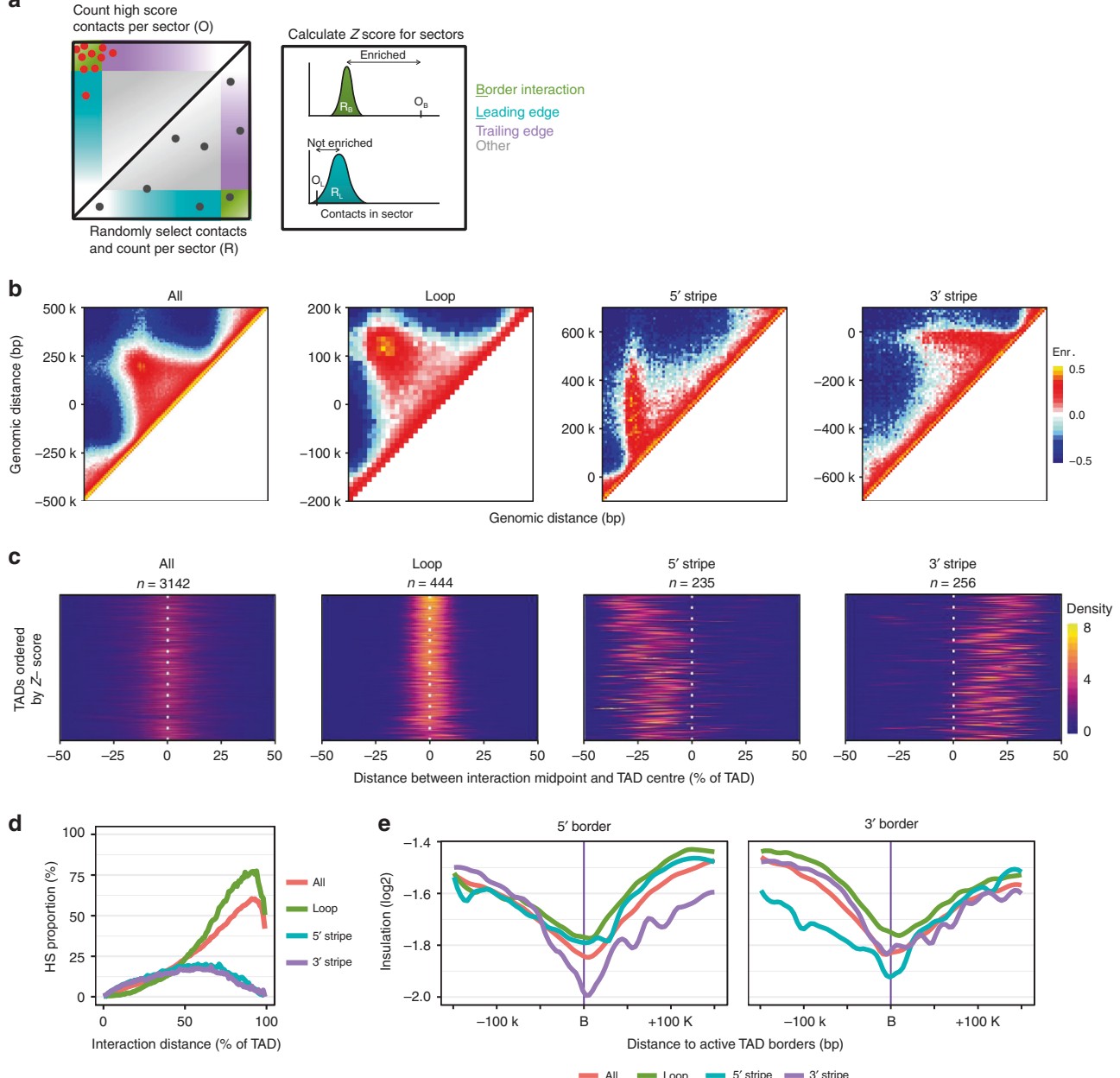

**Fig. 2** Unbiased identification of intra-TAD architecture. **a** Schematic representation of the analysis approach to classify TADs. See the section "Methods" for details. **b** Aggregate Hi-C contact maps of ESC TAD classes identified using the approach in **a**, and size-selected for visualisation ('All', n = 1259; 'Loop', n = 178; '5' Stripe', n = 93; '3' Stripe', n = 102). **c** Distribution of distances between high-scoring contact pairs and the TAD centre, relative to TAD size. All TADs in the class are ordered by Z-score along the y-axis. Yellow represents contact distances with the highest density. **d** Distribution of distances between high-scoring contact pairs in classified ESC TADs. TADs were scaled and discretized into 1% bins with the number of high-scoring interactions (score > 40) in each bin is shown as a proportion of interactions of any score in the same bin. **e** Mean contact insulation (300 kb band) within 150 kb of either the 5' or 3' TAD borders ('B' on the x-axis) and grouped according to TAD class

the borders (Fig. 2b, 'All' panel). 444 loop domains were identified and exhibited high-scoring contact enrichment in the border interaction sector. Of note, loop domains had a comparative depletion of local interactions in the inner-TAD region. Stripe domains, characterised by contact enrichment anchored from one border were stratified into 5'-stripe (n = 235) or 3'-stripe (n = 256) domains based on the location of the anchored border (Fig. 2b, Supplementary Fig. 2a). TADs whose significant sectors included both the corner and leading or trailing edge sectors were also identified in the analysis (Supplementary Fig. 2b, c).

To validate the features associated with stripe architecture, we also classified TADs from various Hi-C datasets including; mouse ESCs grown in two different conditions, 2i and FCS (representing naive or heterogenous pluripotent populations respectively), mouse NSCs, post-mitotic astrocytes (AST)[17], a high-resolution mouse differentiation model[6] and human cells[20]. Stripe TADs were found in all datasets irrespective of differentiation stage (ESC vs. lineage committed), cell cycle distribution, sequencing depth, Hi-C library preparation, TAD segmentation methods and cellular heterogeneity (see results in Supplementary Fig. 2d–j). Collectively, these results support changes to insulation and intra-

TAD connectivity during changes in cell fate and revealed a previously unappreciated heterogeneity in TAD structure in several different cell types, providing us with an opportunity to explore the properties associated with the different TAD classes.

**TAD classes have different internal structure and border behaviour.** To visualise the variability of intra-TAD structures associated with each group, we calculated the distance from the TAD midpoint to the midpoint of the high-scoring contact pair within each TAD (Fig. 2c, Supplementary Fig. 2c, f, "Methods"). This produced a contact distance distribution for each TAD and showed the structural similarities between TADs of the same classification (Fig. 2c). Specifically, the distribution in the loop domain class showed that contact pairs were separated by a distance approximately the size of the TAD (i.e. the midpoint of contact pairs was close to the TAD midpoint) with little variability in the group. On the other hand, the contact distributions in stripe domains were skewed towards the anchored boundary and there was more variability within the class compared to loop domains. This was further revealed when analysing the intra-TAD contact distances between the TAD classes (Fig. 2d, Supplementary Fig. 2g). Loop domains contained a higher proportion of high-scoring interactions that spanned the entire domain and a comparative lack of near-*cis* interactions, as expected if there was a specific loop interaction between the borders. In contrast, stripe domains contained a wide range of contact interaction distances with a broad distribution and had very few interactions covering the entire domain (Fig. 2d, Supplementary Fig. 2g).

The aggregate plots of the stripe domains suggested differences in border behaviour. To quantify this, we calculated the contact insulation around each border of each TAD class. As expected, when 'all TADs' or loop domains were considered, they exhibited a symmetric insulation profile at both their 5′ and 3′ borders (Fig. 2e, Supplementary Fig. 2e, h), indicating that both boundaries were similarly well defined. In contrast, we observed different contact insulation profiles at the unanchored borders of stripe domains. While specific and similar contact insulation is observed at both borders of loop domains, insulation at the unanchored border of a stripe domain (i.e. the 3′ border of a 5′-stripe domain) decays across a region spanning up to 100 kb internal to the TAD (Fig. 2e, compare the blue line between left and right panels, Supplementary Fig. 2e, h). These results highlight both the different border behaviour between TAD classes (loop vs. stripe domains), as well as between the borders of the stripe domain class, suggesting a variability in the choice of terminating border in stripe domains.

Finally, we assessed the relationship between TAD classes and epigenetic properties using publicly available datasets (Supplementary Table 2). In general, stripe domains carried the hallmarks of active genomic regions. In comparison to loop domains, stripes were enriched in the active compartment and were significantly longer, earlier replicating, gene-rich, and had higher levels of gene expression (Supplementary Fig. 2m, n) (KS test, $p < 0.001$). Taken together, the classification of TADs into distinct structural groups emphasises the heterogeneity inherent in TAD structure and has revealed a diversity of TAD border behaviour. Thus, the distinct TAD classes offer an opportunity to explore the mechanisms that contribute to the formation of these different domain structures, their biological functions and their developmental dynamics.

**Both primed and active enhancers at anchored borders of stripe TADs.** Visual inspection of several stripe domains revealed a relationship between the anchored border and enhancer histone marks. For example, genes in the Olfactory receptor (*Olf*) cluster

on chromosome 10 are not expressed in ESCs and are contained within a 5′ stripe domain (Fig. 3a). The previously annotated enhancers, 'Poros' and 'Kithira'[28] are located within the stripe domain. Poros is located at the anchored border of the stripe where it is enriched for H3K4me1 and H3K27me3 marks, but not Pol2, suggesting it may act as a poised enhancer element in these cells (Fig. 3a). Similarly, in NSCs, the Protocadherin beta (*Pcdh-β*) and gamma (*Pcdh-γ*) clusters on chromosome 18 are also contained within a stripe domain, this time anchored from the 3′ end. The annotated enhancer, 'ccr' is known to regulate *Pcdh-β* and *Pcdh-γ* expression, but not *Pcdh-α* expression[29] which resides in a neighbouring TAD. Like in the *Olf* cluster, ccr is located at the anchored border of the stripe, is enriched for the active enhancer marks H3K4me1 and H3K27ac and in this case, makes specific interactions with the -β and -γ genes to activate their expression in NSCs (Fig. 3b).

To explore the relationship between enhancers and the anchored border of stripe domains genome-wide, we determined the average relative position of ChIP-seq profiles for the enhancer histone marks (H3K27ac and H3K4me1), as well as the pluripotency-specific transcription factors (TF) Nanog, Oct4, and Sox2 with respect to the TAD classes in ESCs. TAD classes in the repressive compartment exhibited weaker ChIP signals compared to active compartment TAD classes (Supplementary Fig. 3a), thus we focused our analyses on active compartment TADs. We observed distinct distributions of enhancer marks and TFs among the TAD classes. Loop domains had an overall lower ChIP-seq signal for both enhancer marks, as well as TFs compared to stripe domains (Supplementary Fig. 3a), likely reflecting the overall active state of the stripe domain class. In addition to these signal differences, the distribution of the histone marks and TFs was very different in the stripe domains, where the signal was maximal at the anchored border and progressively decreased throughout the TAD towards the unanchored border (Fig. 3c, Supplementary Fig. 3a, b).

To further validate these results, we used ChromHMM to analyse the distribution of enhancer epigenetic marks and define chromatin states in active compartment TAD classes. Using H3K27ac, H3K27me3, H3K4me1, and Pol2 ChIP data, we considered four enhancer states: Primed, Poised, Active with Pol2, and Active without Pol2 in both ESC and NSC (Fig. 3d, Supplementary Fig. 3d, e Supplementary Table 2). Similar proportions of loop and stripe domains contained enhancers of all classes (Supplementary Fig. 3f). However, the distribution of enhancer elements across TAD classes was non-uniform, with specific enrichment of both Poised and Active chromatin states at the anchored border of stripe domains compared to both the unanchored border and to loop domains (Fig. 3d). Our results indicate that the localisation of poised or active enhancers at the anchored end of a stripe domain is associated with the polarity of stripe domain architecture.

**Enhancers at anchored stripe borders are enriched for Cohesin and NIPBL.** Several studies have reported a link between enhancers, gene transcription, and binding of the cohesin-loading factor NIPBL[30–33]. Given our observations of differential enhancer landscapes between the TAD classes, we reasoned that this may contribute to differential NIPBL-mediated cohesin loading within the TAD classes. We therefore analysed NIPBL and Smc1 ChIP-seq data from ESCs[31] within our classified TADs. Genome-wide NIPBL ChIP-seq signal was higher in TADs in the active compared to the repressive compartment and its distribution across all ESC TADs showed that a higher proportion of NIPBL was located within TADs (Supplementary Fig. 3g), compared to CTCF which was enriched near to TAD borders. As with

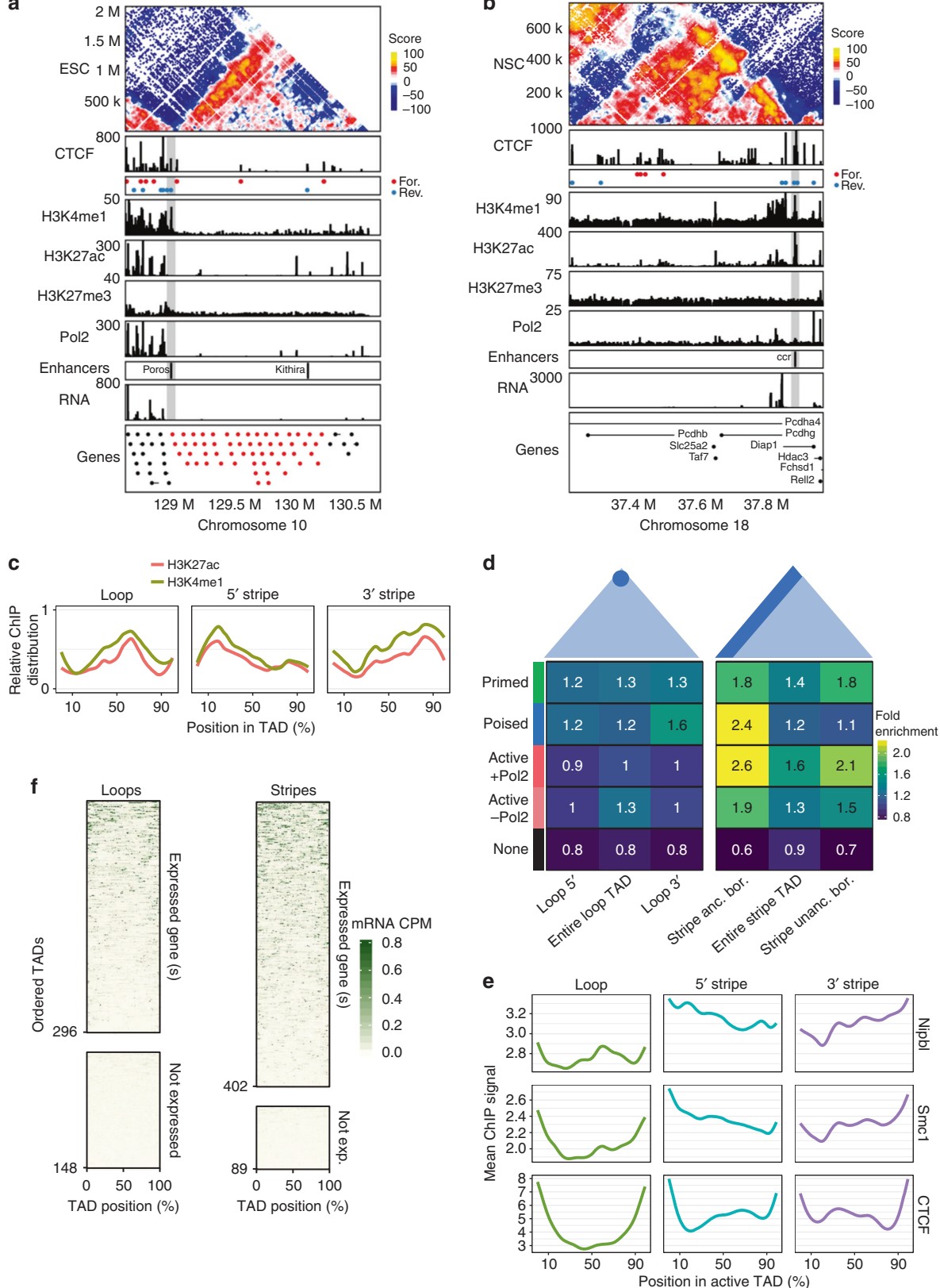

the distribution of enhancer marks and pluripotency TFs (Fig. 3c), the distributions of NIPBL, CTCF, and cohesin across TADs were distinct between TAD classes (Fig. 3e). In active compartment loop domains, there was an overall lower level of NIPBL signal compared to stripe domains and NIPBL was less

enriched at the TAD borders. In stripe domains, NIPBL signal was maximal at the anchored border and progressively decreased throughout the TAD towards the unanchored border (Fig. 3e, top row), as was observed for the enhancer marks and TFs. Interestingly, the relative enrichment of Smc1 followed the same

**Fig. 3** Lineage-appropriate enhancers and cohesin loading, not transcription, define stripe TAD polarity. **a** Hi-C contact map for a 2 Mb region around the Olfactory (Olf) gene cluster on chr10 in ESCs. Also shown is ChIP-seq data (CTCF, H3K4me1, H3K27ac, H3K27me3, Pol2), CTCF motifs at bound sites (annotated as before), RNA-seq data and the locations of the known Olf enhancers Poros and Kithira. Only the TSSs of the genes in the region are shown for clarity, with Olf TSSs coloured in red. Note, the cluster is located at the end of the chromosome, hence the abrupt lack of Hi-C data in the image. **b** Hi-C contact map for a 700 kb region around the Protocadherin (Pcdh) beta and gamma clusters on chr18 in NSCs. Epigenomic tracks as in **a**, including the known Pcdh enhancer ccr. **c** Scaled ChIP-seq signal distributions of enhancer histone marks (H3K4me1, H3K27ac) across active ESC TAD classes. **d** ChromHMM was used to define genomic segments into Primed (H3K4me1), Poised (H3K4me1, H3K27me3), Active (H3K4me1, H3K27ac ± Pol2) and None in ESCs. The enrichment of these segments with respect to the TAD borders of loop (left panel) or stripe domains (right panel) was then calculated using ChromHMM OverlapEnrichment (see also the section "Methods"). Fold enrichment values are shown. Borders were defined as a 60 kb region of the TAD and 5′ and 3′ stripe domains were oriented and grouped into anchored (anc.) or unanchored (unanc.) borders. **e** Distribution of ESC RNA-seq signal (counts per million) in 5 kb bins across scaled loop and stripe TADs calculated using deepTools. TADs were ordered according to the mean expression within the TAD (see the section "Methods" and Fig S3h). **f** Distribution of the mean ChIP-seq signal for Nipbl, Smc1, and CTCF across size-scaled and classified ESC TADs of the A compartment. Note, the signal was not scaled to allow comparison between TAD classes

trajectory as NIPBL across stripe domains (Fig. 3e, middle row). While CTCF distributions in all TAD classes showed enrichment at the borders, there was a reproducible, but mild enrichment at the anchored border of stripe domains compared to the unanchored border (Fig. 3f, bottom panels and also see Fig. 4).

**Transcription is not a key determinant of stripe TAD polarity.** RNA polymerase II tracks along chromatin while transcribing genes, and in some cases the active polymerase can remain associated with the promoter whilst tracking along the gene body[34]. Furthermore, cohesin is reported to be positioned by polymerase during transcription[30,35]. Thus, we examined whether transcription was correlated with the polarity of stripe architecture. RNA-seq data from mouse ESCs[36] was used to determine the distribution and orientation of transcription across all loop or stripe domains. In support of the example regions in Fig. 3, this global analysis revealed that neither gene activity nor the distribution of actively transcribed genes within stripe domains could account for stripe directionality (Fig. 3f, Supplementary Fig. 3h, i). Further, by separately considering forward and reverse RNA-seq alignments, we found no evidence for a correlation between the direction of gene transcription and the polarity of a stripe domain (Supplementary Fig. 3i). Taken together, our results indicate that neither transcription nor its direction are key determinants of stripe domain polarity. Rather, our results support a model whereby NIPBL takes advantage of accessible chromatin at lineage-poised or active enhancers to load cohesin, perhaps leading to asymmetric loop extrusion, and thereby linking lineage-appropriate accessibility with cell-type-specific genome topology.

**Context-specific CTCF occupancy and distribution among TAD classes.** CTCF binding and the orientation of its consensus motif are important determinants of TAD borders[25,27,37–40]. Thus, the localisation and occupancy of convergent CTCF sites would be expected to impact TAD border definition. We used CTCF ChIP-seq datasets from ESCs to define how the CTCF-binding landscape influences TAD class structure. When we visually compared TAD classes with CTCF-binding profiles and motif orientations, we observed that loop domains were associated with prominent, and often clustered, convergent CTCF-binding sites at both domain borders (Fig. 4a). Meanwhile stripe domains had an increased number of CTCF-binding sites throughout the domain and often the strongest signal CTCF site was located at the anchored stripe border (Fig. 4b). These observations were supported with a genome-wide analysis which revealed context-specific CTCF occupancy and distribution differences among the TAD classes. First, there were more CTCF peaks within stripe domains compared to loop domains (KS test,

$p < 0.001$) (Fig. 4c, Supplementary Fig. 4a). Second, the distribution of CTCF-bound motif orientations was correlated with TAD class. When considering all TADs, there was a preference for CTCF to be bound at forward motifs at the 5′ border and reverse motifs at the 3′ border (Fig. 4d), as previously reported. This bias for bound convergent motifs at TAD borders was similar in loop domains. However, while the forward motif preference was observed prominently at the anchored border of a 5′ stripe domain, there was not a strong bias for reverse motifs specifically at the unanchored border, but rather a lower level of motif bias more widely distributed within the domain, suggesting multiple possible reverse-motif partner sites for the anchored CTCF-binding site (Fig. 4d, Supplementary Fig. 4b, c). The complementary trend was observed in 3′ stripe domains. CTCF peaks under which no motif could be identified did not show biased distribution across TAD borders and the same trends were observed in NSCs (Supplementary Fig. 4b, c).

The differential distribution of CTCF signal and the differential bias for motif orientation suggests that TADs, guided by the CTCF landscape, have distinct potentials for structure formation. To further understand the relationship between CTCF sites and domain topologies, we calculated the intra-TAD distance between convergent CTCF pairs that could interact, stratified by TAD class. This revealed that the distances of convergent CTCF peaks in loop domains spanned a larger proportion of the TAD than in stripe domains (Fig. 4e, Supplementary Fig. 4d, median separation 64% compared to 31% for loop and stripe domains, respectively), and indicated that convergent CTCF sites in stripe domains mediate interactions that span shorter distances than the overall size of the TAD.

To understand the strength of interactions associated with convergent CTCF sites in the different domain classes, we analysed the aggregate interaction signal from convergent intra-TAD CTCF pairs in each TAD class. While interactions between convergent CTCF pairs are enriched in all TAD classes, the interactions are strongest in loop domains (Fig. 4f, g). Contact enrichment scores in loop domains were significantly higher than in all ESC TADs (KS test, $p < 0.001$), but were weaker in stripe domains (KS test, $p < 0.001$). CTCF motif entropy could not account for these differences, since it was similar irrespective of TAD class (Supplementary Fig. 4e). Overall, our results reveal that differences in CTCF occupancy, distribution, and interaction potential act in concert to influence TAD architecture.

**Developmental dynamics of TAD architecture.** If stripe domains represent a regulatory architecture underlying cellular identity, then changes to these structures might be expected to be concomitant with differentiation. To explore this, we tracked the dynamics of TAD class accompanying lineage-commitment.

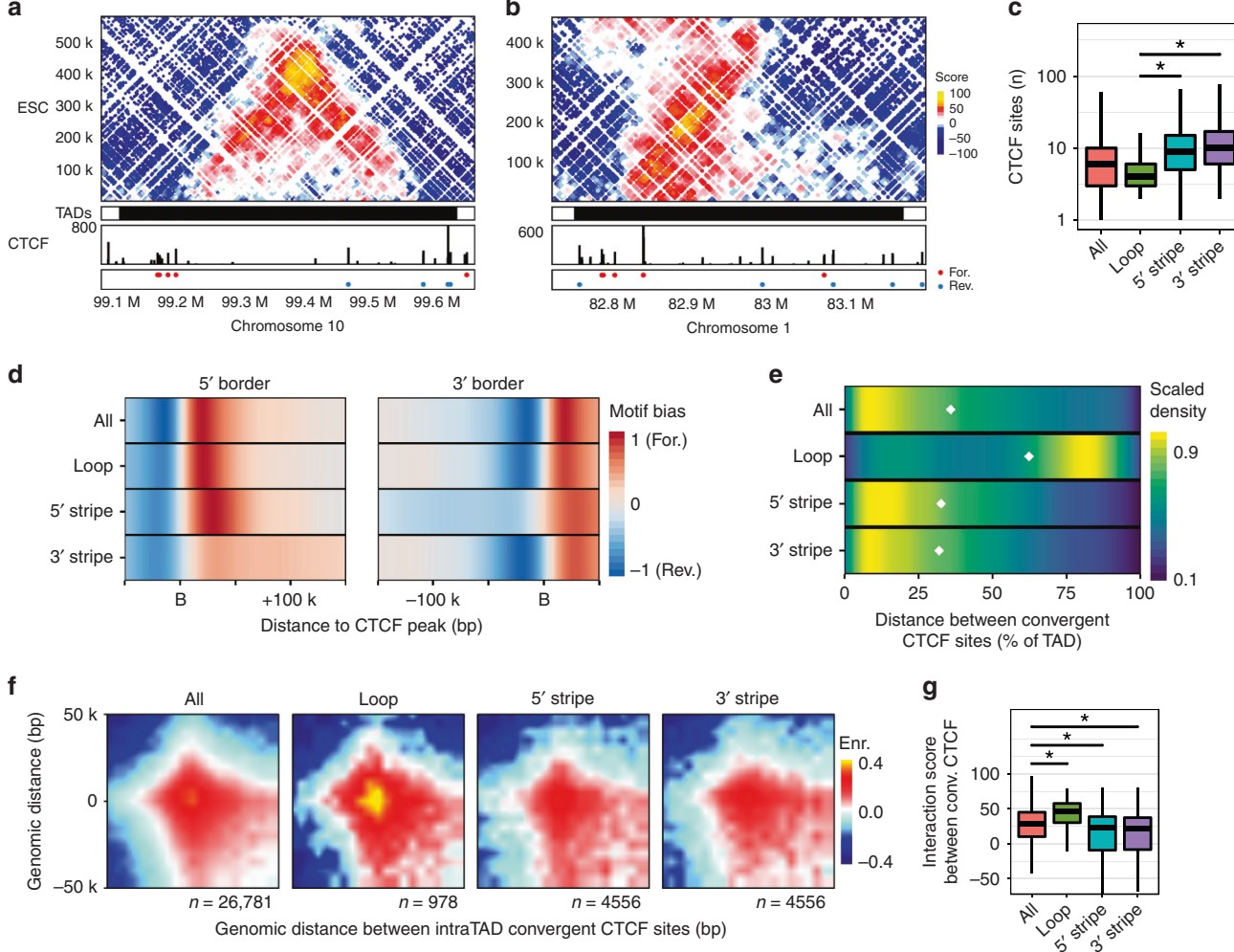

**Fig. 4** CTCF occupancy and connectivity are key determinants of stripe TAD architecture. ESC Hi-C contact maps for (**a**) a 550 kb region on chr10 representative of a loop domain and (**b**) a 450 kb region on chr1 representative of a 5′ stripe domain. TAD calls, CTCF ChIP-seq data, and motifs at peaks are shown as before. **c** Distribution of the number of CTCF-binding sites in classified ESC TADs. Central bar represents the mean and whiskers and boxes indicate all and 50% of values, respectively. KS test *p < 0.001. **d** Distribution of bound CTCF motifs within 50 kb outside and 100 kb inside of TAD borders, separated into 5′ and 3′ borders ('B' on the x-axis). Colour scale represents the difference in density distribution of forward and reverse motifs. **e** Density distribution of distances between convergent CTCF motifs within TAD classes as a proportion of TAD size. White diamonds indicate median distances. **f** Aggregate Hi-C contact maps around pairs of intra-TAD convergent CTCF sites in ESCs. Pairs have a minimum interaction score of 40 within a 50 kb region. Number of pairs shown below each plot. **g** Quantification of interaction scores between convergent CTCF sites shown in (**f**), Box plot as in **c**, KS test *p < 0.001

Given the fact that 5′ and 3′ stripe TADs are likely fundamentally the same type of structure and the fact that we did not expect that a change from a 5′ to a 3′ stripe TAD would represent a change in TAD class, we combined 5′ and 3′ stripe TADs into one group. We first compared the tendency for TAD classes to be associated with the active or repressive compartment with differentiation. We observed an overall bias for TADs to become repressed during differentiation (more active TADs switch to repressive TADs), in line with published reports[6]. Both loop and stripe domains had a similar behaviour, albeit a smaller proportion of stripe domains switched compartment between ESC and NSC (Supplementary Fig. 5a).

To investigate whether TADs switched classes during differentiation, we compared TAD classes, where at least one border was maintained between the cell types. This revealed that loop domains were more stable through differentiation than stripes, where 61.5% of NSC loop domains arose from loops in ESCs, compared to 37.5% of NSC stripe domains (Fig. 5a, b, grey).

We considered stripe dynamics in three groups: 'emergent', 'maintained', and 'lost' stripes. Several interesting features can be observed. First, NSC stripes are either 'maintained' from ESC stripes or they newly 'emerge' during differentiation. Second, the 'emergence' of NSC stripes from ESC loops happens more frequently (19.8%, green) than the 'loss' of ESC stripes to NSC loops (10.5%, blue) (Fig. 5a, b). Our results argue that stripe domains are more susceptible than loop domains to structural change during differentiation and provided us with an opportunity to explore the mechanisms contributing to these changes.

The formation of a stripe domain is associated with the occupancy and arrangement of CTCF sites (Fig. 4) and lineage-appropriate enhancer-coupled cohesin loading (Fig. 3). Thus, we reasoned that changes to CTCF-binding site occupancy or enhancer activity, may contribute to changes in TAD connectivity and class. Given the marked reduction in CTCF occupancy at the unanchored borders of stripe domains (Fig. 4), we investigated the impact of cell-type-specific changes in CTCF occupancy on

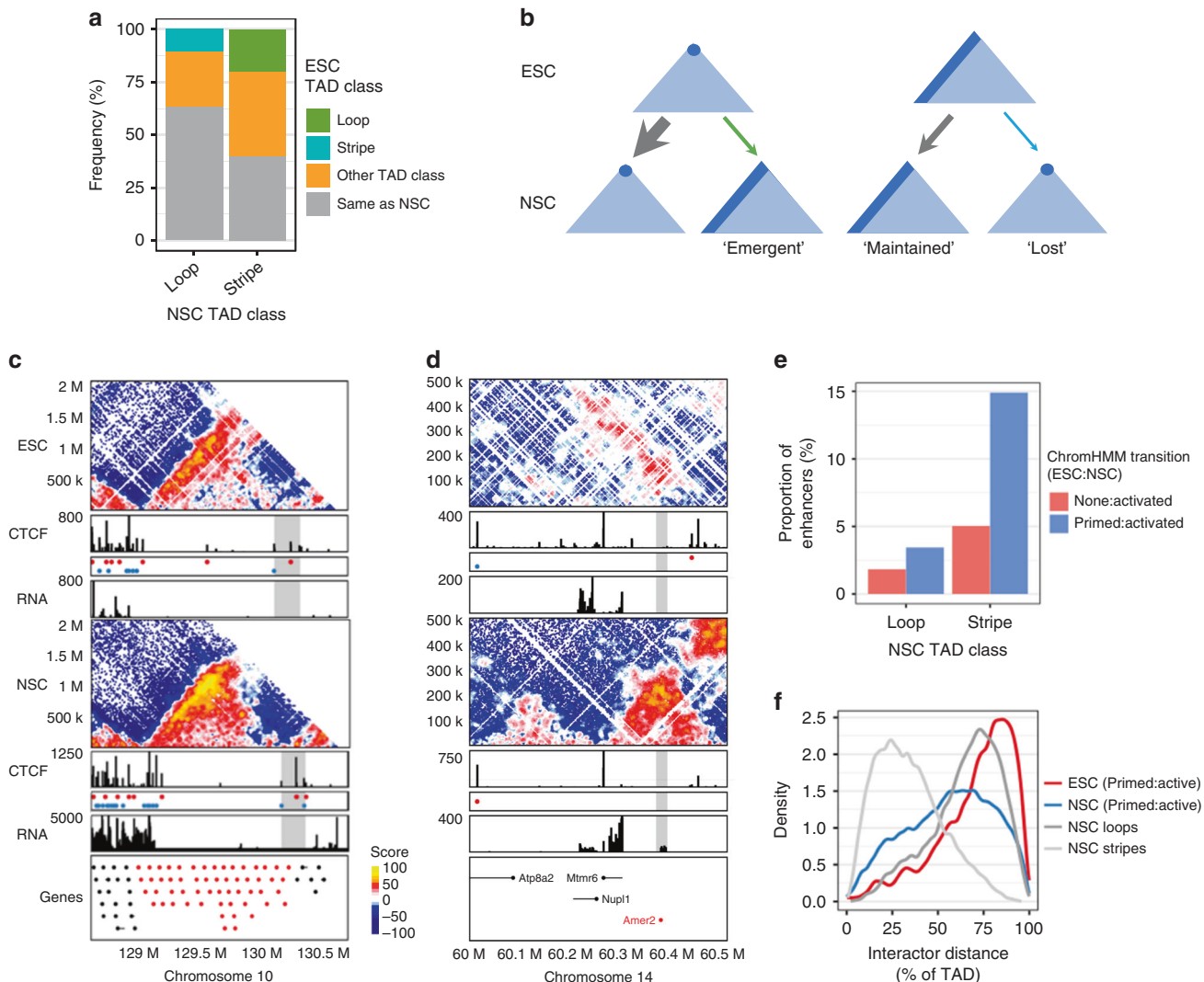

**Fig. 5** Structural changes to stripe TADs accompanying differentiation are coincident with changes in CTCF occupancy or lineage-appropriate enhancers. **a** TAD class dynamics between ESC and NSCs. ESC TADs separated into loop or stripe domains that maintain at least one border during differentiation, and their classification in NSCs. Frequency represents the proportion of NSC classified domains that were maintained or changed from a given ESC TAD group. **b** Schematic representation of data in (**a**) revealing the fates of TAD classes during differentiation. **c**, **d** Hi-C contact maps from ESCs and NSCs revealing TAD class changes associated with CTCF-binding site occupancy or transcription changes. CTCF ChIP-seq tracks, motifs, gene annotations, and gene expression are shown as before. **c** A 2.1 Mb region on chr10 around the Olf gene cluster changes from an ESC stripe to an NSC loop domain in conjunction with an increase in CTCF signal in NSCs at the unanchored stripe border (highlighted grey). **d** New insulation at a 500 kb region on chr14 is associated with a change from an ESC stripe to an NSC loop domain upon NSC-specific expression of *Amer2*. **e** Activated NSC enhancers defined by ChromHMM which were either unclassified ('None', red) or Primed (blue) in ESCs and their distribution in NSC loop or stripe domains. **f** Distribution of intra-TAD contact distance as a proportion of TAD size, in ESC (red) or NSC (blue) Hi-C maps from active NSC-specific enhancers which were primed in ESCs (Primed:active). As a comparison, the distribution of contact distances in loop (dark grey) or stripe (light grey) NSC domains are also shown. Note the shift between the blue and red distributions upon activation of NSC enhancers. Enhancer elements were expanded by 20 kb to include enough contact information

TAD class change in the context of an ESC stripe domain that is 'lost' in NSCs (Fig. 5a). For example, in ESCs the olfactory gene cluster is contained within a stripe domain whose anchored border is enriched for enhancer marks and unanchored border contains a cluster of low signal CTCF-binding sites. Upon differentiation, the occupancy of a CTCF site at the unanchored stripe domain border is increased, concomitant with a change in TAD class to a loop domain in NSCs (Fig. 5c). Importantly, we note that stripe domains are also observed to be 'lost' in association with lineage-specific gene expression and in the absence of changes to CTCF occupancy (Fig. 5d, Supplementary Fig. 5b). For example, *Amer2* is a negative regulator of the Wnt

pathway and is expressed in NSC. It is located within a stripe domain in ESCs where it is not expressed. The stripe domain is 'lost' in NSCs and a new insulation site is observed, which is independent of CTCF occupancy changes but associated with NSC-specific expression of *Amer2* (Fig. 5d).

As enhancer marks were enriched at the anchored border of stripe domains (Fig. 3), we investigated whether the 'maintained' or 'emergent' stripe classes were associated with lineage-specific enhancer activation. We identified ESC-specific or NSC-specific enhancer elements using ChromHMM analysis (Fig. 3, Supplementary Fig. 3e). Enhancers which become activated in a cell-type-specific manner were more enriched in stripe domains in

that cell type. For example, 19.9% of activated enhancer transitions were in NSC stripes compared to 4.8% in NSC loops (Fig. 5e). Activated NSC enhancers arise from both 'primed' as well as 'none' chromatin states in ESCs, with the former being more strongly enriched (Fig. 5e, Supplementary Fig. 5c). Interestingly, poised NSC enhancers are also enriched in NSC stripes, in agreement with the enrichment of poised enhancers at the anchored border of ESC stripe domains (Supplementary Figs. 5c, 3d). Conversely, a larger proportion of NSC stripe domains contain NSC-specific active enhancers compared to loop domains, although this was specific for the active enhancers which arose from already primed states in ESCs (Supplementary Fig. 5d).

Using this information, we explored whether lineage-specific enhancer activation contributed to the 'emergence' of stripe conformations. For example, an ESC loop domain on chromosome 12 contains primed chromatin states throughout the TAD. In NSCs, the TAD is associated with a stripe conformation and activated enhancer marks appear towards the anchored border of the stripe domain (Supplementary Fig. 5d, e). To explore this globally, we took advantage of the fact that loop and stripe domains have characteristic intra-TAD contact distributions (Fig. 2d). We reasoned that if the activation of an enhancer element in NSCs contributed to a 'stripe' conformation, then the contact distribution profile would contain a higher frequency of short-range interactions as was characteristic for stripe domains. We identified the chromatin segments that were associated with primed–active transitions in ESC–NSC, and calculated the contact distribution profile specifically from these sites. We chose primed–active transitions because primed chromatin states were not observed to be associated with stripes in ESCs (Fig. 3d). Activation of NSC-specific enhancers was associated with a shift in the contact distribution profile in NSCs (Fig. 5f, compare red to blue line) towards the distribution observed in stripe domains (Fig. 5f, compare blue to light grey line), suggesting that lineage-specific enhancer activation can contribute to the unique topology of stripe domains.

## Discussion

To provide insight into how topological chromatin configurations relate to cellular potential, we classified TAD structural heterogeneity and quantified the characteristics of TAD boundary behaviour during differentiation. We observe quantitative changes to the TAD landscape accompanying the loss of pluripotency and the acquisition of the differentiated state. Topological restriction during lineage commitment is reflected in terms of changes to insulation, border behaviour, and the nature of intra-TAD connectivity. These results help to reconcile the observation of chromatin plasticity in ESCs[41] despite similar TAD numbers[4], are in keeping with reports of genome structure constraints accompanying cell specification[5,6] and with the increased definition and decreased dynamics of chromatin domains accompanying embryoid body differentiation using microscopy[42].

Our observations of changes to intra-TAD connectivity and border behaviour between cell types led us to develop an unbiased method for the identification and characterisation of intra-TAD connectivity, and ask whether these structures represent a regulatory architecture underlying cellular identity. It is known that many developmental enhancers act not only upon their target gene, but also on unrelated neighbouring genes within the same environment[43]. Furthermore, lineage-specific enhancers can 'track' along chromatin scanning for their biological promoter[44]. Such events may be predicted to have a unique chromatin structure, which resolves into a 'line' or 'stripe' of interactions in population-based Hi-C maps. Like commonly described 'loop

domains', 'stripe' structures are readily identified in Hi-C contact maps, yet are often overlooked in analyses due to the paucity of methods to identify them in an unbiased manner. While we were preparing our study, a similar stripe architecture was described[24]. The methods used by Vian et al.[24] to define stripe TAD structure differ from those described herein. Where their method does not necessitate TAD definition and stripes are defined through manual curation, our analysis involves segmentation into TADs followed by unbiased classification. In this context, some stripes, such as those that extend through neighbouring TADs, may be missed if the TAD segmentation method does not capture nested TADs[45–47]. Similarly, stripe TADs share properties with previously described FIREs[48], in that they represent a near-diagonal Hi-C hotspot of local chromatin interactions, however FIREs are not necessarily at TAD borders. Finally, while 5′ or 3′ stripe domains have different relative anchored borders, it is likely that they represent the same TAD structural class. In this context, it may be preferable to consider stripe domains as having a 'leading' or 'trailing' edge.

Vian et al.[24] describe a similar enrichment of cohesin and NIPBL at enhancers located at anchored stripe borders suggesting that both methods capture similar architectural features. Here we show that apart from simply being structural features of genomes, stripe domain formation is linked to the functional state of the cell. The presence of enhancers at the anchored end of stripe domains and the nature of interactions emerging from these sites throughout the domain strongly supports the notion that stripe domains represent enhancer-scanning tracks in the genome. Importantly, as both silent and active landscapes are contained within stripe domains, we show that transcription is not a key determinant of stripe conformation. Thus, stripe architecture could provide the 3D framework for both productive as well as opportunistic enhancer–promoter tracking, or possibly serve as a rich environment for the emergence of new regulatory elements[2] or disease states[49].

There may be multiple mechanisms of TAD formation that resolve into a stripe domain using C-based methods. We propose that cohesin loading takes advantage of chromatin accessibility at lineage-poised or active enhancers and from here initiates TAD formation, whether that be by loop extrusion[22,23], or other dynamic mechanisms. Indeed, it may be possible to simulate stripe TAD structure using asymmetric extrusion models where the border at the leading edge would be anchored while the border at the trailing edge could engage more freely in interactions through the genome. We note that all TADs may in fact be formed through an asymmetric process but only resolve into stripes when CTCF occupancy is compromised. By hijacking accessible chromatin to facilitate cohesin loading, productive lineage-appropriate enhancer–promoter interactions could be reinforced and thus cell-type-specific gene programs would be supported coincident with TAD formation.

A factor defining the specificity of enhancer–promoter interactions is spatial co-localisation[2,50,51] and when insulators are compromised, enhancer-mediated interactions have scope to reach beyond the TAD and spread in cis into adjacent domains[11]. By regulating CTCF occupancy, the effective TAD boundary position becomes variable, allowing enhancers to reach genes located beyond or in adjacent TADs. Indeed, we observe a differential distribution and occupancy of the CTCF-binding landscape in stripe domains compared to canonical TADs. The unanchored border of stripe domains are associated with lower CTCF occupancy and the lack of a dominant convergently bound site. In this context, multiple convergent CTCF sites could potentially act as the interactor at the trailing edge of the TAD which may introduce cell-to-cell variability and lead to a diversity of terminating boundaries within the sampled population.

Mechanisms that impact the occupancy of CTCF, such as motif methylation[52], or its rapid chromatin dynamics[53], perhaps influenced by local disruptive chromatin activities, might result in a poorly defined boundary in the population. Furthermore, developmental changes to CTCF occupancy impact TAD border definition, suggesting that developmentally regulated mechanisms that lead to low occupancy CTCF sites in specific regions of the genome may facilitate permissive topologies and regulatory opportunities.

Finally, while we did observe cases of TAD border definition coupled to changes in CTCF occupancy, overall the correlation between occupancy and insulation was not absolute, similar to recent observations[5]. Strongly bound CTCF sites did not always exhibit insulation and new insulation which arose during development was not always associated with changes in CTCF occupancy. Additional mechanisms may contribute to CTCF's insulation functions. In flies, numerous insulator proteins cooperate to bring about context-specific insulation at TAD borders[54] and recent reports in mammalian cells have implicated YY1[16,55] as a player in genome organisation with CTCF. Thus, the heterogeneity of TAD structure may, in part, involve novel insulator proteins. Whether mammalian cells exhibit insulator diversity and whether such diversity scales with topological changes during development are important questions for future research.

Our results show that CTCF and lineage-specific enhancers are key features promoting cell-type-specific genome topologies. We propose that these structures permit regulatory plasticity and facilitate the formation of lineage-specific regulatory networks during cell fate transitions.

## Methods

**Embryonic stem cell culture and Hi-C library preparation.** Mouse ESC were cultured in media supplemented with fresh Leukaemia Inhibitory factor (LIF) and the two inhibitors, PD0325901 (MEK inhibitor) and CHIR99021 (GSK-3 inhibitor) (Stemgent) to maintain them in the groundstate of pluripotency and passaged at regular intervals. The ESCs used in this study were derived in the laboratory of Duncan Odom (CRUK CI) and routinely cultured in 2i conditions in-house. ESC Hi-C libraries were prepared from 10 million cells as previously described[17] with the following modifications. After accutase treatment and washing, cells were passed through a 40 μm strainer and fixed in wash media with 1% formaldehyde (FA) for 10 min at room temperature (RT). FA was quenched using 0.125 M glycine for 5 min at RT. Cells were washed in 10 ml of cold PBS twice, centrifuging the sample at 750 rcf at 4 °C between washes. Cell pellets were resuspended in 10 ml Hi-C lysis buffer (10 mM Tris–HCl pH 8.0, 10 mM NaCl, 0.25% NP-40, 1x EDTA-free protein inhibitors (Roche) and incubated on ice for 25-30 min. The cell lysis was aided by 10 strokes in a glass douncer using 'tight' pestle and nuclei spun down at 800 rcf at 4 °C for 5 min. Nuclear pellets were resuspended in 500 μl 1.2x NEB2.1 (New England Biolabs (NEB)) with 0.3% SDS, transferred to low-retention tubes (Eppendorf Protein Lo-Bind) and incubated at 37 °C for 30 min in a thermomixer (950 rpm). The temperature was then adjusted to 60 °C and nuclei were incubated at 60 °C for 5 min. Triton X-100 was added to the nuclei to final concentration of 2% and nuclei incubated for 15 min at RT with tilting to quench SDS. 1500 U HindIII (NEB) was added to the samples to digest the chromatin in nucleo. After 4 h at 37 °C, 750 U of fresh enzyme along with 10 μl NEB10x were added to the samples and incubated overnight at 37 °C (650 rpm). In the morning, the nuclei were pelleted for 5 min at 4 °C and 800 rcf, resuspended in 500 μl 1X NEBuffer 2.1, and chromatin further digested using 1000 U HindIII for 4 h at 37 °C shaking at 650 rpm and occasionally resuspended using low-retention tips. Filling of digested ends was performed as previously described. Nuclei were washed twice in 100 μl 1X T4 DNA ligase buffer (NEB) and resuspended in 200 μl of the same buffer. Ligation was performed in three steps: (1) 6 μl of T4 DNA Ligase (NEB M0202) was added to the nuclei and incubated at 16 °C for several hours with occasional resuspending; (2) 10 μl 10X buffer and 4 μl Ligase were added and samples incubated at 16 °C overnight; (3) nuclei were pelleted, resuspended in 100 μl fresh 1X ligase buffer, 6 μl ligase added, and incubated at 16 °C for 4 h with occasional resuspending. Biotin clean-up and sonication were performed as previously described, with the exception that the sonication time was shortened to 70 s to get average fragment size ~200 bp and thereby increase the proportion of informative fragments in the libraries after size selection. Size selection was done using SPRI Size Selection Beads (Beckman Coulter). To remove fragments <150 bp, 1.1X of SPRI beads was used (120 μl of sonicated DNA with 132 μl beads). After incubation at RT for several minutes, samples were put on a magnet. Short fragments are retained in the supernatant which is removed after 5 min incubation on the magnet. Beads were washed twice in 85% EtOH, dried and libraries eluted in 100 μl nuclease-free water

after 3 min on the magnet. Biotin pull-down and library preparation were performed as previously described. Libraries were prepared from 15 to 20 μg DNA in 5 μg aliquots.

**ChIP-seq analysis.** Public datasets used in this study are listed in Supplementary Table S2). ChIP-seq dataset quality was verified using FastQC to visualise base call quality and ensure the sequencing was free of contaminants. Libraries were aligned to the mm10 reference genome using Bowtie 1.1.2 in -v alignment mode and converted to BAM format. Up to three mismatches were permitted across the read and a unique alignment was required (-v 3 -m 1 —best). Samtools was then used to sort the maps and remove potential PCR duplicates. Alignment maps were imported into R and extended by 150nt to represent the sonication fragment and binned at 50 bp resolution. Peaks of binding were then identified by transforming normalised read density ($v$) using $-\log_{10}(1-\mathrm{quantile}(v))$, as previously described in refs. [17,25].

**ChIP-seq profiles across genomic points.** The ChIP signal within the specified distance of a genomic feature was extracted from the binned ChIP-seq profile (created from the aligned ChIP-seq data as described above). The distance of the ChIP signal to the genomic feature was binned at 1 kb resolution and the mean ChIP signal within the 1 kb bins calculated.

**ChIP-seq profiles across scaled regions.** Alignment frequency across the genomic regions was extracted from the alignment map and the distance of a ChIP signal to the start of the genomic region calculated. The distance was then converted to a proportion of the genomic region and binned to two decimal places. Within the genomic region, the mean signal within a bin was calculated to produce a mean signal profile for each region. The profiles for each region were smoothed using Loess to produce a profile for a ChIP-seq track across many genomic regions.

**CTCF motif identification.** Peaks of CTCF binding were identified (at a log-percentile threshold of 3, see above) and the genomic sequence underlying the peak was queried using FIMO[56] for the CTCF motif(s) with a $q$-value threshold of 1%. Where one peak contained multiple significant motifs, the most significant was considered as representative of the peak.

**ChromHMM genome segmentation.** Alignment maps for IP and Input datasets for ESC or NSC cell types were submitted to ChromHMM for segmentation using a bin size of 200 and an extension size of 150 bp. All other parameters were defaults. Each IP target was represented for each cell type at least once: H3K4me1, H3K27ac, H3K27me3, and Pol2 (see Supplementary Table 2 and Supplementary Fig. 3d). Additionally, a state with no association to any included ChIP-seq dataset was termed 'None'. This analysis was later expanded to include many more marks and states, which were all assessed for enrichment at borders of classified TADs (Supplementary Fig. 3).

**Summary of Hi-C library processing.**

1. To produce the Hi-C libraries reported in this study we employed 'in situ Hi-C' as previously reported[17,25] and further described below.
2. All Hi-C data reported was generated using Illumina paired-end sequencing. These are represented as 'Total Sequenced Reads' in Supplementary Table 1.
3. Hi-C data was processed using a custom pipeline as previously reported[17,25] and which was further optimised for parallel computation on a cluster. Sequenced reads were aligned to the mm10 genome (further details below), 'Uniquely mapped Reads' in Supplementary Table 1.
4. Read pairs are then filtered to remove ambiguous (chimeric) or unaligned reads. This results in the 'Total Contact Pairs' in Supplementary Table 1.
5. Construction of contact matrices. Hi-C matrices are created at the fragment end (f-end) level by assigning each read to a f-end. The reads used at this stage are from 'Total Contact Pairs' in Supplementary Table 1. Additional filtering steps occur at this stage including: reads that are not sufficiently close to a f-end and contact pairs that are from self-ligation events.
6. Analysis of Hi-C library Quality Control includes, *cis:trans* ratios, f-end coverage (Supplementary Fig. 1a–c), Hi-C library replicate correlations (Supplementary Fig. 1d) and contact probability distributions (Supplementary Fig. 1e, f).
7. Normalisation of contact matrices. We use a method termed 'Shaman' developed in the lab of Amos Tanay[26]. More details are included below.

**Hi-C sequence alignment and quality control.** The quality of sequenced reads was verified using FastQC. Although base call quality scores were confirmed to be high throughout the cycles, there was no spatial bias in read quality, and no sequences were overrepresented. Paired-end reads were aligned independently to the mm10 reference genome using Bowtie 1.1.2 in -v alignment mode. Unique alignments were required and up to three mismatches permitted (-v 3 -m 1 --best). Once aligned, the read pairs were reunited to filter out reads where either (or both)

failed to align. Additionally, read pairs that were aligned in very close proximity were removed. Finally, potential PCR duplicates were removed by comparing the aligned coordinates of read pairs in each map. Maps for biological and sequencing replicates were merged once the quality of the replicates had been validated (using the Hi-C data processing steps below). Sequencing replicate maps were merged and the PCR duplicate removal step applied to the composite map. The composite maps were used for all downstream analyses. Hi-C read statistics for ESC, AST, and NSC datasets are shown in Supplementary Table 1.

**Hi-C data processing and normalisation.** Genomic coordinates of read pair alignments were attributed to HindIII restriction fragment ends and near-cis interactors removed as previously described[17,25]. The number of aligned reads, number of reads associated to a fragment end, and the number of fragment ends identified can be found in Supplementary Fig. 1 and Supplementary Table 1. To retain maximum information at all scales, analyses were performed at fragment end resolution with no binning unless otherwise indicated. We used a non-parametric, bin-free analytical approach termed SHAMAN to normalise Hi-C data, and determine interaction scores. The Shaman R package was developed by Amos Tanay. For specific details, please refer to ref. [26]. Briefly, the method first randomises Hi-C matrices so to preserve both genomic distance and marginal contact distributions but specific features, such as compartments, TADs, or loops are not maintained. The randomisation algorithm repeatedly samples pairs of observed contacts and shuffles them (using a Markov Chain Monte Carlo-like approach) by swapping the fragment end points with a ratio proportional to their contact distance probabilities, correcting for the asymmetric distribution of sampled and shuffled distances (please see their manuscript for further information). We note that this method controls for marginal coverage effects. Observed contact densities are then compared to the randomised contact densities, generating a normalised interaction score (as referred herein). Specifically, to calculate the interaction score for each observed contact, the distribution of Euclidean distances to the nearest $k$ neighbours was determined in the observed ($k = 100$) and shuffled ($k = 200$) matrices and then compared using the Kolmogorov–Smirnov $D$ statistic to visualise positive (higher density in observed data) and negative (lower density in observed data). These $D$-scores are used to visualise interactions on a $-100$ to $+100$ scale, and referred to as *Interaction scores* in the text.

**TAD identification and TAD boundary calling.** TAD borders for ESC datasets were identified as previously described[25] using interacting fends separated by between 100 and 400 kb. Borders were defined as a 1 kb region about the maxima within the regions in the top 95th percentile of the scaling track. TADs were then defined as the genomic regions between these TAD borders. In addition, TAD borders were identified using the observed insulation track of a 250 kb region to identify genomic regions where insulation was above the top 15th percentile (as in refs. [6,57]). Within these regions, the minimal insulation position was used to define the TAD border and TADs were filtered to be between 100 kb and 4 Mb. TAD maps were subsequently categorised into two self-interacting groups and the LaminB1 track used to predict the group being the active or repressive compartment. ESC or NSC LaminB1 tracks were downloaded from UCSC and the mm9 coordinates converted to mm10 using liftOver.

**Contact insulation.** A one-dimensional track of contact insulation was calculated from the observed fragment end interaction maps as described[6,57]. Unless otherwise stated, the insulation tracks were computed at a 1 kb resolution within a 50 or 300 kb region to identify localised or broader insulation points, respectively. Very near *cis* interactions within the 0–10 kb band (for the 50 kb region) or 0–20 kb band (for the 300 kb region) were not considered. Insulation was plotted either as a distribution of mean contact insulation 200 kb around borders (Fig. 1c) or as a distribution of insulation at border positions (Fig. 1d).

**TAD connectivity analysis.** This analysis was performed as described in ref. [5]. For a given TAD, the high-score contacts (>40) were found where one fragment end was within the TAD and the other was also in the TAD or up to 10 Mb from either TAD border. Contacts were then categorised as intra-TAD if both fragment ends were within the TAD borders or inter-TAD if one of the fragment ends was located in the upstream or downstream region. The proportion of intra-TAD high-score contacts to the total number of high-score contacts (intra-TAD + inter-TAD) was calculated.

**Aggregate Hi-C maps.** To visualise the enrichment of Hi-C contacts over many distinct genomic loci, the observed and shuffled contact matrices were compared. At each genomic region and bin size specified, the number of contacts in the binned matrix was summed. The summary matrix was then normalised to the maximal value, the observed/shuffled calculated and visualised on a log2-transformed scale. For aggregate analyses that compare whole TADs, the set of TADs visualised are selected so that the size of the TADs considered are most similar. Unless stated, the resolution of these plots is 5 kb.

**Classification of TAD structures.** To identify distinct distributions of interacting contacts within genomic regions, we developed a method termed 'domainClassifyR' github.com/ChristopherBarrington/domainClassifyR. The method requires two inputs: (1) 2D regions within which contacts should be classified and (2) locations of contacts and an associated score from which 'high-scoring contacts' (score of >40) can be filtered (Supplementary Fig. 6a). TADs were filtered to have a minimum size of 200 kb before classification. In the schematic shown in Supplementary Fig. 6b, the contacts within the 2D regions defined by A:A and B:B would be considered; this would classify TADs A and B individually. The 2D regions to be classified need not necessarily be TADs therefore, rather these are regions of the genome that contain observed contacts. Each of the 2D genomic regions are discretised into four sectors: corner (C), leading edge (L), trailing edge (T), and other (O). These sectors are used to attribute a contact to a type of interaction (Supplementary Fig. 6c). The size of the edge also determines the size of the corner, for this study the size of the edge is set to 60 kb. The corner of a TAD therefore represents a 60 kb$^2$ region and the edges represent a ~60 kb*(TAD length$-60$ kb) region. The 60 kb edge size parameter is influenced by dataset size; this parameter describes how far from a TAD border an interaction can be that could be considered as spanning the entire TAD.

The contacts that are wholly contained within the bounds of the 2D region are then grouped into one of the four sectors. The contacts that meet the minimum score threshold are then identified; these contacts are the 'high-scoring' contacts from 'all' contacts. For this analysis, a contact is deemed 'high-scoring' when its score is more than 40. A null distribution of contact positions is then generated; this describes the number of contacts that may be expected to be in a given sector if there was no bias due to interaction score. The set of all contacts is sampled $L$ times (in this study, $L = 1000$), selecting $N$ contacts randomly (using the $R$ sample() function), where $N$ is the number of high-scoring contacts in the 2D region. At each iteration, the number of randomly selected contacts in each sector is calculated. A table of $L$ rows and four columns is produced, where each column describes the null distribution for a sector in the 2D region. The number of high-scoring contacts in each sector is then computed. For example, a 2D region is defined and is found to contain 5000 observed contacts, of which 900 are above the minimum score threshold and 600 are positioned in the corner sector. Within this domain, 900 of the 5000 contacts are randomly selected and the distribution among sectors calculated. This is repeated 1000 times to generate the null distributions. For each sector, the number of observed high-scoring contacts is compared to the distribution of randomly selected contacts using the $Z$ statistic. Conversion of the $Z$-score to a $p$-value and Benjamini–Hochberg correction for multiple testing (across all 2D regions) converts the $Z$-score into a $Q$-value. A pair of thresholds is applied to the $Z$-scores: a minimum threshold to identify sectors that are enriched and a maximum threshold to identify sectors that are not enriched. Sectors with $Z$-scores between these thresholds may be considered ambiguous and are excluded from further analysis with this thresholding approach. In this study, a sector must score more than 15 to be enriched, between 5 and 15 to be ambiguous, and <5 to be not enriched.

The enrichment of each sector in 2D regions is calculated independently so each region may have multiple enriched sectors. The classification of the 2D region is the aggregate of the enriched domain sectors. Regions that have a sector classified as ambiguous are excluded from further analysis. Increasing the lower threshold (from 5) or decreasing the upper threshold (from 15) would decrease the number of ambiguous sectors identified but could decrease specificity (or homogeneity) of the classified groups. Therefore, a TAD presented here with a 'Corner' classification (loop domain) shows specific (and exclusive) enrichment of high-scoring contacts in the 60 kb interacting region between the TAD borders. TADs with multiple enriched sectors were not considered; in these TADs, the Leading + Corner or Trailing + Corner sectors were most common.

**Comparison of TADs between datasets.** The precise location of TAD borders between multiple datasets is challenging to assess. We note that smaller TAD border windows (i.e. 1 or 10 kb) reduces the number of overlapping TAD borders between cell types. For example, when a 1 kb distance was observed for border overlap between ESC and NSC, this would result in 627 of 4344 ESC borders (14.4%). The overlap between TAD borders increased as the border distance increased; with a 10 kb distance, 1192 of 4344 ESC borders (27.4%) overlapped; with a 50 kb distance, 2997 of 4344 ESC borders (69.4%) overlapped. We proceeded with the 50 kb constraint to be able to have enough common borders to be used downstream. For either dataset, consecutive common borders were then identified. Where a consecutive pair of borders were common to both datasets, the domain was considered to be 'maintained'. Depending on the comparison, the TAD flanked by a consecutive border pair may be 'split' into multiple TADs or be the result of multiple TADs 'merging' together (see schematic in Supplementary Fig. 6d).

**Calculating Fend distance distributions.** For contacts within each TAD, the midpoint between the contacts was calculated and the distance of the midpoint to the centre of the TAD was calculated (Supplementary Fig. 6e). For a contact that was positioned in the corner sector of the domain, the midpoint of the contact would be close to the midpoint of the TAD, so the distance between the midpoints would be small. For a contact in the leading edge sector, the midpoint would be 5′

of the TAD centre, so the distance would be negative. Conversely the midpoint of a trailing edge contact would be 3′ of the TAD centre so the distance would be positive. The distance metric was normalised by TAD size, thereby defining the limits of the distance to between −50% and 50% (% of the TAD). The R density() function was used to plot the frequency distribution of the normalised distance metric. Since the number of TADs in each classification is different, the space occupied by each TAD on the x-axis is variable; each horizontal strip represents a TAD.

**Distance from border to CTCF peaks**. CTCF-binding sites within −50 and 150 kb of a classified TAD border were identified. The genomic distance between the TAD border and the midpoint of the CTCF peak was calculated and the R density() function was used to compute the frequency distribution of TAD-peak distance within TAD classes, border proximity and motif orientation. The distributions were normalised to their maximum values and the reverse motif distribution subtracted from the forward motif distribution; a value close to 1 therefore indicates a high value in the forward motif distribution and a near zero value in the reverse motif.

**Reporting summary**. Further information on research design is available in the Nature Research Reporting Summary linked to this article.

## Data availability
The data generated and analysed in this study have been deposited in the GEO database without restriction and with the accession number SRA668328. Publically available datasets analysed in this study have been included in Supplementary Table 2. Any additional relevant data is available from the authors upon request. All other relevant data supporting the key findings of this study are available within the article and its Supplementary Information files or from the corresponding author upon reasonable request. A reporting summary for this Article is available as a Supplementary Information file.

## Code availability
The code used to run domainClassifyR can be found at github.com/ChristopherBarrington/domainClassifyR.

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

## Acknowledgements

The authors thank members of the Hadjur laboratory for discussions and the Genomics Core Facility in the Cancer Institute for sequencing. This work was supported by the Wellcome Trust (106985/Z/15/Z) granted to S.H.

## Author contributions

C.B. conceived the project, performed the analysis, interpreted the results, and contributed to writing of the manuscript. D.G. performed experiments. D.P. prepared the ESC Hi-C libraries and contributed to the writing of the manuscript. W.V. performed analysis and contributed to the writing of the manuscript. J.H. contributed analysis advice. S.H. conceived the project, interpreted the results and wrote the manuscript.

## Additional information

**Competing interests:** The authors declare no competing interests.

