## [Peer Review File · Nature Communications]

Reviewers' comments:

Reviewer #1 (Remarks to the Author):

In this manuscript, Barrington and colleagues describe their analysis of multiple Hi-C datasets, showing that asymmetric TADs, with one-sided flare-like Hi-C maps are more common with differentiated cells. The authors suggest that these flares (or "stripes") are often associated with poised or active enhancers, as well as CTCF and cohesin sites, and might play a crucial role in the tissue-specific activation of genes. The model they propose is that while topological domains are often conserved, their insulation and symmetry may differ between stem cells and differentiated cells, resulting with an overall different contact maps.

Overall, the paper is clear and will be of great interest to the community. Nonetheless, I had some technical comments, as well as some issues that need to be evaluated in a more systematic and critical manner.

- The authors reach their conclusions based on Hi-C libraries from mouse ESC (presented here), neuronal stem cells and astrocytes (as previously published by their own group - Sofueva et al, and from Tanay's and Cavalli's groups - Bonev et al. To gain confidence, I believe that the authors should take a more systematic approach and analyze multiple Hi-C datasets that are available for both mouse and human. It would be irresponsible to state something as general as "We observed that topological restriction during lineage commitment is reflected in terms of changes to insulation, border behaviour and the nature of intra-TAD connectivity." without checking (available!) data in multiple conditions. Moreover, Hi-C in ESC and during neuronal differentiation have been published before. Are the same phenomena present in those datasets and boundaries as well?

- In addition, since there are so many ways to segment the genome into topological domains, I would appreciate if the authors would use some additional software suites for analyzing Hi-C data, cementing their biological insights and making sure they are not due to some artifact that is more subtle in one condition compared to another.

Some additional comments:

- The title is not informative and could better match the content of the manuscript. Specifically, I recommend that it would refer to the notion of asymmetric TADs and one-sided boundaries that produce the flare-like patterns that are so easy to observe.

- Throughout the paper, the authors tend to infer causality, even when it is unsubstantiated. Please be more cautious.

- I believe that the flares (or stripes) in one-sided TADs could be simulated using an extrusion model whereas one of the sides has been "stuck" while the other side of the loop is "scanning" through the genome. Perhaps the authors should elaborate on this model a bit more.

- Recently, Nora et al (PMID 28525758) published a series of Hi-C and 5C experiments in mouse ESCs during targeted degradation of CTCF. It could be of great interest to analyze how the flare boundaries depend on CTCF, compared to other boundaries.

- I urge the authors to consider how their results may be interpreted in light of a population-view of Hi-C data. Perhaps what we see is not an enhancer "scanning" through the TAD, but just a set of cells that have a similar boundary on one side, but different ones on the other.

- I was surprised to see p-values expressed as percents (e.g. "KS test, $p < 0.1\%$ " line 84).

- Lines 84-86 read: "Altogether, the results indicate that despite similar TAD organisation between the cell types, contact frequencies between and inside TADs are markedly different as cells change fate.". Yet, I wish the authors would quantify these results and assign some statistical confidence by multiple ways. As I said earlier, it was not shown that these "stripes" are not due to other confounding effects (such as cell heterogeneity, # cells, depth, chromatin state, accessibility, library preparation, etc).

- Line 106, please fix unformatted reference to {MedelsonCohen:2017ga} (which actually should be "Mendelson Cohen").

- Overall, the description of Hi-C preprocessing was not clear enough. What is the "interaction score"?, "high-scoring contacts"? How were intra-TAD contacts randomly sampled? The number of Hi-C contacts depend greatly on the interaction distance.

- I'm not sure that the notation of 5'-stripes 3'-stripes is correct, as technically, either side of the TAD is both 5' and 3'. I would prefer left and right, or even "leading" and "trailing" edges, as the ends were later referred to.
- Also, in lines 124-125 the authors report on how many asymmetric TADs they find for each pattern, and I wonder how many are expected by random, using some shuffling model.
- If I understood correctly, the authors see difference is the assignment of "Loop" vs. "Stripe" domains into A and B compartments. If so, I would prefer that they control for these differences when comparing various statistics later on (such as the average CHIP-seq signal of CTCF or NIPBL).
- In lines 257-259 the authors suggest that "our results support a model whereby NIPBL takes advantage of accessible chromatin at lineage-poised or active enhancers to load cohesin". Are there indeed difference in the DNA accessibility between loop and stripe boundaries? This can be easily tested either in the lab or from published ATAC-seq and DHS data in some cell types.
- Line 277, "distribution of CTCF-bound motif orientations was dependent on TAD class". Perhaps rephrasing this as "correlated" would be clearer. Is CTCF binding causal or just correlative with either type of TAD boundaries
- Are more provocative question - is there a continuum of boundary efficiency, where "Loop" domains are just domains with two dominant and fixated boundaries (leading to fewer interactions within), then the regular TADs, and then the "stripe" domains that have one boundary that is fixed and one that is just a little bit weaker or flexible?
- Lines 330-332. Could you please generate "deepTools"-like metaplots/heatmaps of CTCF binding at all ESC-vs-NSC flare boundaries that are gained/maintained/lost, as well as the other boundary for control. This could perhaps test the instructive role of CTCF in forming those domains.
- Lines 324-5. "Maintained" vs "Emergent": this is how they were defined, right?
- Perhaps the authors could perform an unbiased enrichment test for (1) various chromatin marks, (2) binding of various factors, and (3) sequence specific motifs at fixed vs dynamic TAD boundaries, to account for the mechanisms that underlie this fabulous phenomenon they describe.

Reviewer #2 (Remarks to the Author):

Barrington et al. present an analysis that classifies TADs based on details of their contact enrichments. Overall the results are of interest to the field of genome organization, but details of methods and comparisons to other approaches should be addressed before consideration for publication.

Methods:

How were the Kolmogorov-Smirnov D-statistic scores scaled to the interval (-100,100)? What was the original interval of scores before rescaling? The KS statistic is typically the supremum of the absolute value of the difference between two distributions. How does "Enr." (e.g. in Fig1b,d) relate to "Score" (e.g. in Fig1a,e)?

How does the power to detect 5' stripes, 3' stripes, or loops scale with the size of the TAD? The authors could address this by running domainClassify on simulated datasets.

The reported minimum TAD size was 100kb, and the reported width of the edge used was 60kb— this should completely encompass small TADs— how was this dealt with? It would be useful to indicate the filtering of TAD sizes to (100kb, 4Mb) in the main text.

How was maintenance of stripe status determined (Fig5a) if TADs were considered to have split or merged upon differentiation (Methods Fig1)?

What do the distributions of high-scoring contacts within called TADs look like? How were thresholds determined, e.g. what quantile of the distribution is 40? It would be useful to add figures illustrating the various choices of cutoffs, and their underlying distributions, to the methods section.

Results and Discussion:

Related to Fig1d, it is unclear if this change between ESCs and NSCs is due to: a difference in the

internal organization of the TADs, a difference in the size of TADs, or a difference in the overall distribution of D-scores. To address this it would be useful to include the following in the supplement: (i) a histogram of detected TAD sizes for each dataset, (ii) average contact maps stratified by TAD size, (iii) plot of the variance in the D-scores as a function of genomic distance in ESC and NSC (up to the size of largest called TADs). It would be useful to have average plots stratified by domain size for the categories in Fig2b added to the supplement as well.

The fraction of 'loop domains' reported here, 444/3142 is much lower than ~50% reported in Rao et al., 2014. How similar are the 'loop domains' identified here to the previous definition? Was the previous estimate an overestimate? What are some examples of genomic loci where this method would call a 'loop domain' and the other would not (or vice versa)? The authors could address this either by running the previous method on their datasets, or running their method on the Rao et al., 2014 dataset. If the overlap is low, a different terminology may be better suited to describe the domains reported here (e.g. 'corner-enriched').

Relatedly, it is not clear whether it will be more or less confusing in the long run to adopt the same terminology ('stripe domains') as Vian et al., 2018, since the methods for identifying stripes differ. For example, the method here relies on first calling TADs, whereas that one does not. This may have advantages (the method in Vian et al. appeared to require lots of manual curation) or disadvantages (false negatives for stripes that extend through called TADs). Indeed, the authors should address the ambiguity of any particular segmentation into TADs (e.g. Forcato et al., 2017, DOI: 10.1038/nmeth.4325) and how this might impact the method and results described here. Similarly, it could be useful to clarify the difference with FIREs described in Schmidt et al., 2016 (DOI: 10.1016/j.celrep.2016.10.061), and/or discuss the synergies and differences with other decomposition approaches for near-diagonal Hi-C interaction patterns (e.g. Zhan et al., 2017, DOI: 10.1101/gr.212803.116, Norton et al., 2018, DOI: 10.1038/nmeth.4560).

 Small points and suggestions:

-- Fig1a: It would very be useful to provide more information about the normalization method used to produce maps (i.e. SHAMAN with k=100 nearest-neighbors), as it is not widely used, as well as the restriction enzyme used for Hi-C (this can help explain the empty bins). It would also be useful to write out the algorithm name in the text when it is cited for the first time.

-- Fig5a: it is unclear if swapping from a 5' to 3' stripe would be changing class or not, given the wording.

-- FigS5 appears to have a missing panel or an issue with labeling, and a reference appears to be unformatted in the second results section; depositing the manuscript to a preprint server could allow such details to be corrected more quickly.

-- The authors write that "the majority of TAD borders were similar (69.4%, separated by less than 50 kb) between pluripotent and lineage committed cells". It would be interesting to comment more on this in the discussion: is 50kb an experimental limitation (e.g. derived from shifts in boundaries ascertained in different replicates) or do some of these represent biologically important shifts in boundary positioning?

-- the use of 'contact' to interchangeably describe raw contacts from sequencing as well as high-scoring contact pairs after the SHAMAN procedure leads to some confusing moments in the text. Introducing a term for the latter may improve clarity.

-- the code for running domainClassify does not appear to be referred to in the main text or methods. Also, the description of the method is fairly dense, it could help if certain methods or scripts were pointed to in the methods section at the relevant points.

NCOMMS-18-32431 - Response to Reviewers' comments.

Reviewer #1

We would like to thank the reviewer for their support and for the useful critique. As described in detail below, we have incorporated new data to address the comments and believe the manuscript to be significantly improved as a result.

1. The authors reach their conclusions based on Hi-C libraries from mouse ESC (presented here), neuronal stem cells and astrocytes (as previously published by their own group - Sofueva et al, and from Tanay's and Cavalli's groups). To gain confidence, I believe that the authors should take a more systematic approach and analyze multiple Hi-C datasets that are available for both mouse and human. It would be irresponsible to state something as general as "We observed that topological restriction during lineage commitment is reflected in terms of changes to insulation, border behaviour and the nature of intra-TAD connectivity." without checking (available!) data in multiple conditions. Moreover, Hi-C in ESC and during neuronal differentiation have been published before. Are the same phenomena present in those datasets and boundaries as well?

We thank the reviewer for this comment. We have now expanded our analysis of topological restriction during lineage commitment as well as confirming the features associated with stripe architecture in human cell lines (HCT116), mouse ESCs grown in two different media conditions (and therefore representing different functional states of pluripotency and different levels of functional heterogeneity), post-mitotic astrocytes (AST), and in a high-resolution cellular differentiation model (ESC/NPC/CN from Bonev Cell 2017). Collectively, the additional results support changes to insulation and intra-TAD connectivity during changes in cell fate (please see additional data, Fig S1m-u). Our results are in line with several reports that describe quantitative differences in genome topology during developmental transitions (Bonev Cell 2017) and reprogramming (Stadhouders Nat Gen 2018). Furthermore, when we process these datasets through domainClassifyR, we obtain TAD structures with similar border behaviours as was shown in the manuscript (please see Fig S2d-h). Thus, we believe that i) our analysis supports the existing ideas of topological restriction during lineage commitment and ii) our ability to identify stripes and their properties exists irrespective of differences in cell number, heterogeneity, sequencing depth and library preparation. We have indicated these supporting data in the manuscript (Lines 87-92 and Lines 133-140).

2. In addition, since there are so many ways to segment the genome into topological domains, I would appreciate if the authors would use some additional software suites for analyzing Hi-C data, cementing their biological insights and making sure they are not due to some artifact that is more subtle in one condition compared to another.

We have now used two additional TAD segmentation methods on two different datasets and observe similar results. (Fig S2i,j). We do however note that as our method involves classification after TAD calling, some stripes, such as those that extend through neighbouring TADs, may be missed if the TAD segmentation tool used does not capture nested TADs. This comment has been included in the discussion (Lines 416-418).

3. The title is not informative and could better match the content of the manuscript. Specifically, I recommend that it would refer to the notion of asymmetric TADs and one-sided boundaries that produce the flare-like patterns that are so easy to observe.

We have now adjusted the title.

4. I believe that the flares (or stripes) in one-sided TADs could be simulated using a extrusion models whereas one of the sides has be "stuck" while the other side of the loop is "scanning" through the genome. Perhaps the authors should elaborate on this model a bit more.

We agree that this is a possible model and have now added a sentence to the discussion to highlight this possibility (Lines 441-443).

5. Recently, Nora et al (PMID 28525758) published a series of Hi-C and 5C experiments in mouse ESCs during targeted degradation of CTCF. It could be of great interest to analyze how the flare boundaries depend on CTCF, compared to other boundaries.

We agree with the reviewer that this is indeed an interesting question. We have downloaded the suggested Hi-C data and processed it through our pipelines and domainClassifyR in an attempt to assess the role of CTCF in stripe architecture. As before, we were able to identify stripes from this additional ESC (control) dataset. However, it was difficult to assess the impact on stripe structure in CTCF-AID cells (Fig S4f) since very few TADs from the mutant Hi-C contained enough data to be classified. From those which could be classified, there appears to be little/no change in stripe architecture in the absence of CTCF. Please note, in order to do the analysis shown in S4f, we had to reduce the minimum number of high-scoring interactions/TAD, remove the score threshold and plot all profiles irrespective of TAD size into the aggregate plots. Thus, we do not feel this is a very robust analysis. A complete assessment of the role of CTCF in stripe architecture will require significantly more input (ie. targeting either specific CTCF sites, or specific TADs). This is beyond the scope of this manuscript however it is a very interesting direction for further study which we hope to address in future.

6. I urge the authors to consider how their results may be interpreted in light of a population-view of Hi-C data. Perhaps what we see is not an enhancer "scanning" through the TAD, but just a set of cells that have a similar boundary on one side, but different ones on the other.

We agree with the reviewer. Multiple convergent CTCF sites could potentially act as the interactor at the trailing edge of the TAD which may introduce cell-to-cell variability and lead to diversity of terminating boundaries within the sampled population. We believe that our results support this hypothesis. This would best be addressed using scHi-C data and we did attempt to do this before submitting the manuscript for review, however due to the low resolution of these datasets, we were unable to robustly show this. We have further highlighted these ideas in the discussion (Line 457-461).

7. I was surprised to see p-values expressed as percents (e.g. "KS test, $p < 0.1\%$ " line 84).

These have been changed to $p = 0.001$.

8. Lines 84-86 read: "Altogether, the results indicate that despite similar TAD organisation between the cell types, contact frequencies between and inside TADs are markedly different as cells change fate.". Yet, I wish the authors would quantify these results and assign some statistical confidence by multiple ways.

We believe the reviewer is referring to the data shown in Fig 1. In Fig 1b/c, we examined insulation (inter-TAD contacts) in ESC compared to NSC in two ways i) Aggregate Hi-C maps of contacts across all TAD borders (Fig 1b) and ii) the distribution of mean insulation across the borders of all

TADs (Fig 1c) shown at multiple different distance scales (Fig S1i). As per the reviewers request, we have now included an analysis of *TAD Insulation* (Fig 1d) quantified according to the methods published in Bonev et al and Stadhouders et al (see also Methods Lines 136-138) and revealing a statistically significant difference.

Furthermore, in Fig 1, we examined TAD *connectivity* (intra-TAD contacts) in two ways; i) Again we show an aggregate Hi-C map of matched, size-selected TADs in the two cell types and then quantify the distribution of interaction scores between the borders of the TAD (Fig 1e,f). This analysis did reveal a statistically significant difference in border contact between the cell types. This was supported by an analysis of contacts specifically between intra-TAD convergent CTCF sites (Fig S1j). In response to the reviewers request, we have now also included an analysis of TAD *Connectivity* (Fig 1g), quantified according to the methods published in Stadhouders et al. In this analysis, connectivity is defined as the number of high-score contacts (>40) that are located within a TAD as a proportion of high-score contacts that connect the TAD to a 10Mb region up or downstream of the TAD (included in the Methods, Lines 150-156).

The above analyses have now also been performed in ESCs grown in 2i or serum conditions which represent different cell states of functional pluripotency, where 2i ESCs have a more 'naïve' identity (Fig S1m-q). Here too, we observe quantitative changes to topology in line with changes in cell fate.

9. *As I said earlier, it was not shown that these "stripes" are not due to other confounding effects (such as cell heterogeneity, # cells, depth, chromatin state, accessibility, library preparation, etc).*

Please see response to question no. 1 above.

10. *Line 106, please fix unformatted reference to {MedelsonCohen:2017ga} (which actually should be "Mendelson Cohen").*

This is now corrected, thanks for noticing.

11. *Overall, the description of Hi-C pre-processing was not clear enough. What is the "interaction score"?, "high-scoring contacts"? How were intra-TAD contacts randomly sampled? The number of Hi-C contacts depend greatly on the interaction distance.*

We apologize that the reviewer feels this was not clearly presented. We have included further details for the normalization and scoring methods, as well as the classification of TAD structures in the Methods section. Please note, the SHAMAN method used in this study to normalise and score Hi-C data was developed in the laboratory of Amos Tanay (reference included in manuscript) and provided to us through collaboration. We have now more clearly referred to their BioRxiv manuscript as well as expanding the description in the Methods (Lines 115-131). Specifically, to address the reviewers questions; i) SHAMAN involves randomization of Hi-C matrices followed by a comparison of the observed contact densities with the randomized contact densities, generating a normalized D-score/interaction score. ii) We considered interactions with a score of >40 to be 'high-scoring'. iii) We have expanded our explanation of how random sampling of intra-TAD contacts was performed. Please see the Methods and response to point 13 below.

12. *I'm not sure that the notation of 5'-stripes 3'-stripes is correct, as technically, either side of the TAD is both 5' and 3'. I would prefer left and right, or even "leading" and "trailing" edges, as the ends were later referred to.*

We have included a comment about the fact that 5' and 3' stripes are likely fundamentally the same type of TAD structure in the manuscript and could be better considered as a TAD with a leading or trailing edge (Lines 420-423).

13. Also, in lines 124-125 the authors report on how many asymmetric TADs they find for each pattern, and I wonder how many are expected by random, using some shuffling model.

The SHAMAN method (developed by Amos Tanay) first randomizes Hi-C matrices so to preserve both genomic distance and marginal contact distributions. Briefly, they developed a Markov Chain Monte Carlo randomization algorithm by repeatedly sampling pairs of contacts and shuffling them by swapping the fragment end points with a ratio proportional to their contact distance probabilities, correcting for the asymmetric distribution of sampled and shuffled distances (please see their manuscript for further information). Observed contact densities are then compared (in a bin-free manner) to the randomized contact densities, generating a normalized D-score, which is what we used in domainClassify to identify TAD classes. The shuffled/randomized Hi-C matrices do not reveal features of stripe TADs. Thus, the chance that a stripe TAD is observed by random has been considered using this method.

14. If I understood correctly, the authors see difference is the assignment of "Loop" vs. "Stripe" domains into A and B compartments. If so, I would prefer that they control for these differences when comparing various statistics later on (such as the average ChIP-seq signal of CTCF or NIPBL).

This is an important point and we have indeed tried to control for this. The analyses shown in Fig 2e; Fig S2m,n; Fig 3c,e and Fig S3a-c were done in A compartment classified TADs. The reviewer is correct that this was not clearly described in the manuscript or the figure legends and we have now added this information throughout (various lines).

15. In lines 257-259 the authors suggest that "our results support a model whereby NIPBL takes advantage of accessible chromatin at lineage-poised or active enhancers to load cohesin". Are there indeed difference in the DNA accessibility between loop and stripe boundaries? This can be easily tested either in the lab or from published ATAC-seq and DHS data in some cell types.

We have analysed DNA accessibility using DNase I ESC ChIP-seq ENCODE data. This shows an enrichment of DNase I signal at the leading edge of a stripe domain (Fig S3c).

16. Line 277, "distribution of CTCF-bound motif orientations was dependent on TAD class". Perhaps rephrasing this as "correlated" would be clearer. Is CTCF binding causal or just correlative with either type of TAD boundaries.

We have changed this to correlated.

17. Are more provocative question - is there a continuum of boundary efficiency, where "Loop" domains are just domains with two dominant and fixated boundaries (leading to fewer interactions within), then the regular TADs, and then the "stripe" domains that have one boundary that is fixed and one that is just a little bit weaker or flexible?

We thank the reviewer for this point. Indeed, this is how we have interpreted our results as well.

18. Lines 330-332. Could you please generate "deepTools"-like metaplots/heatmaps of CTCF binding at all ESC-vs-NSC flare boundaries that are gained/maintained/lost, as well as the other boundary for control. This could perhaps test the instructive role of CTCF in forming those domains.

We agree with the reviewer that this is an interesting analysis. Based on our results from Fig 4, we believe that changes to lineage-specific CTCF occupancy is associated with transitions in TAD class between cell types. While we were able to identify specific examples of this (ie. Fig 5c), we were not able to observe this trend with confidence globally. There are several reasons for this.

First, as shown in Fig 4 and discussed by the reviewer, multiple convergent CTCF sites could potentially act as the interactor at the trailing edge of the TAD, which may introduce cell-to-cell variability and lead to diversity of terminating boundaries within the sampled population. While interesting, this posed a problem for our global analysis. Closer examination of several specific TADs changing class revealed that there were often multiple (sometimes clustered) CTCF sites at the trailing edge of stripe TADs and that often one (not all) of these changed occupancy. Once such signals had been averaged within a genomic bin for analysis in deepTools, these locus-specific changes in occupancy were lost.

Second, there were few TADs contained within the gained/maintained/lost category, limiting the strength of the analysis. Finally, it was not always the CTCF site at the trailing edge of the TAD which changed. Rather, we observed that a significant fraction of stripe TADs (53%) were neighbored at their trailing border by an anchored TAD border (207 of 387 stripes, $p < 0.001$, chi-squared test). By comparison, only 20 stripe TADs were neighbored by another stripe TAD oriented in the opposite way. This suggests that an anchored border from a neighbouring TAD may function to limit the extent of border definition in stripe TADs. In this context, the occupancy of the CTCF site at the neighbouring TAD border would be defining the limit of the trailing stripe TAD border.

Thus, like question 5 above, we are very interested to further explore the relationship between CTCF occupancy and insulation in the future, where we could take advantage of specific genomic regions identified herein and target specific CTCF binding sites using CRISPR methods to determine the instructive role of CTCF and its occupancy.

19. Lines 324-5. "Maintained" vs "Emergent": this is how they were defined, right?

That is correct.

20. Perhaps the authors could perform an unbiased enrichment test for (1) various chromatin marks, (2) binding of various factors, and (3) sequence specific motifs at fixed vs dynamic TAD boundaries, to account for the mechanisms that underlie this fabulous phenomenon they describe.

In response to the reviewers request we have now expanded our ChromHMM analysis to include a wide array of epigenetic features (Fig S3j).

Reviewer #2

We would like to thank the reviewer for their support and for providing detailed and constructive suggestions. Following the comments, we have clarified the text and Methods, expanded the manuscript with new supporting data and compared our method with previously published approaches more extensively.

1. Methods: How were the Kolomogorov-Smirnov D-statistic scores scaled to the interval (-100,100)? What was the original interval of scores before rescaling? The KS statistic is typically the supremum of the absolute value of the difference between two distributions. How does “Enr.” (e.g. in Fig1b,d) relate to “Score” (e.g. in Fig1a,e)?

We apologize that this was not sufficiently clear. We have now expanded the Methods section to further clarify that the SHAMAN methods used for Hi-C normalization and interaction scores and that these were developed by Amos Tanay and his team and a manuscript is currently under review (Lines 115-131 in Methods). Unscaled D-scores range from 0-1. Enrichment values and interaction scores are different analyses. Enrichment values represent observed contacts in a bin over shuffled while SHAMAN interaction scores represent observed contact densities compared to the randomized contact densities in a bin-free manner.

2. How does the power to detect 5' stripes, 3' stripes, or loops scale with the size of the TAD? The authors could address this by running domainClassify on simulated datasets.

Thank you for this important question. We agree that smaller stripe domains may be missed out in our analysis due to the minimum TAD and border size parameters we have used. Indeed, the question could be addressed by running domainClassifyR on simulated datasets, however my team does not have the expertise to perform or interpret this kind of analysis. In an attempt to address the reviewers comment, we selected a few stripe domains from our analysis in ESCs and computationally adjusted the trailing edge border in 0.5Mb steps. At each border step, we re-ran

domainClassifyR on the adjusted data to assess whether the method was able to identify stripes (Response Figure 1). This revealed that while some TADs are classed as ‘ambiguous’ (because classification is based in the proportion of reads within the TAD), overall however, domainClassifyR correctly classifies the increasingly smaller TADs as stripes as well.

Response Figure 1. The ‘trailing’ (3’) border of a large forward-stripe classified ESC TAD was progressively brought closer to the ‘leading’ (5’) border in 500kb steps (indicated by horizontal lines). Each sub-TAD was then classified using domainClassifyR using the same parameters as in the Methods. The resulting classification for each sub-TAD is shown on the right of the image where ‘FOR’ represents a forward stripe TAD and ‘AMB’ represents a TAD classed as ambiguous (see Methods).

3. The reported minimum TAD size was 100kb, and the reported width of the edge used was 60kb—this should completely encompass small TADs— how was this dealt with?

Apologies if this was confusing. The minimum TAD size for the classification was 200kb. Indeed, this information was missing in the Methods and has now been added (Methods Line 169).

4. *It would be useful to indicate the filtering of TAD sizes to (100kb, 4Mb) in the main text.*

This has been further clarified in the Methods (Lines 138-139). The Observed Insulation method to define TADs (as was described in Nagano et al) filtered for TAD sizes between 100kb and 4Mb.

5. *How was maintenance of stripe status determined (Fig5a) if TADs were considered to have split or merged upon differentiation (Methods Fig1)?*

Thanks for raising this, it may not have been clear. Figure 5a accounts for merged and split TADs by focusing on maintained borders from which a loop/stripe is observed in ESC. This was stated in lines 328-329 and we have now further included this in the legend (Line 587 in Fig 5a legend).

6. *What do the distributions of high-scoring contacts within called TADs look like? How were thresholds determined, e.g. what quantile of the distribution is 40? It would be useful to add figures illustrating the various choices of cutoffs, and their underlying distributions, to the methods section.*

We have now included a distribution of contacts within called TADs into the Methods section (Methods Figure 1). Scores of 40+ represent the 93.5th percentile in our ESC dataset.

9. Results and Discussion: *Related to Fig1d, it is unclear if this change between ESCs and NSCs is due to: a difference in the internal organization of the TADs, a difference in the size of TADs, or a difference in the overall distribution of D-scores. To address this it would be useful to include the following in the supplement: (i) a histogram of detected TAD sizes for each dataset, (ii) average contact maps stratified by TAD size, (iii) plot of the variance in the D-scores as a function of genomic distance in ESC and NSC (up to the size of largest called TADs). iv) It would be useful to have average plots stratified by domain size for the categories in Fig2b added to the supplement as well.*

We have now included data to address the above queries in the supplemental figures. i) Fig S1k, ii) Fig S2a, iii) Fig S1l, iv) please see the result in ii).

10. *The fraction of 'loop domains' reported here, 444/3142 is much lower than ~50% reported in Rao et al., 2014. How similar are the 'loop domains' identified here to the previous definition? Was the previous estimate an overestimate? What are some examples of genomic loci where this method would call a 'loop domain' and the other would not (or vice versa)? The authors could address this either by running the previous method on their datasets, or running their method on the Rao et al., 2014 dataset. If the overlap is low, a different terminology may be better suited to describe the domains reported here (e.g. 'corner-enriched').*

Thank you for this important question. We agree that it is important to clarify terminology and whether our loop domains resemble the loops already described in the literature. The aim of our analysis was to define and characterize stripe domains. As domainClassifyR was not setup to compare our corner-enriched 'loop' domains with currently described 'loops', our parameters are not designed to 'maximise' the number of loop domains in the data. Nevertheless, we downloaded HCT116 Hi-C data (from Rao Cell 2017), analysed it using our pipeline and processed it with domainClassifyR with the aim of comparing Rao loops (defined using HICUPS in their analysis) with our loop domains. Rao et al identified 3,170 HCT116 HICUPS loops, with the majority in the

domain-size range 100kb-1Mb (average size 365kb). Using our aggregate analysis pipelines, we find that these HICCUP loops exhibit a strong 'peak profile' as expected (please see Fig S3k).

The proportion of HCT116 classified TADs (using domainClassifyR) that intersect a HICCUPS-annotated loop was high (Fig S3l), irrespective of the Z-score thresholds used. This is expected since it is known that a loop is usually associated with the appearance of a domain demarcated by the loop (Rao 2014). Meanwhile, a small proportion of HICCUPS loops intersected a 'loop domain' annotated by domainClassifyR (Fig S3l). This was expected since due at least in part to the minimum threshold of TAD size (200kb) that we apply and which would exclude many loops from this comparison. Thus, our results support the fact that the 'loop domains' presented here share structural characteristics with HICCUPS loops (ie. their borders form a chromatin loop), but they represent a subset of all loops. In this context, we feel it is reasonable to maintain the 'loop domain' terminology.

11. Relatedly, it is not clear whether it will be more or less confusing in the long run to adopt the same terminology ('stripe domains') as Vian et al., 2018, since the methods for identifying stripes differ. For example, the method here relies on first calling TADs, whereas that one does not. This may have advantages (the method in Vian et al. appeared to require lots of manual curation) or disadvantages (false negatives for stripes that extend through called TADs). Indeed, the authors should address the ambiguity of any particular segmentation into TADs (e.g. Forcato et al., 2017, DOI:10.1038/nmeth.4325) and how this might impact the method and results described here. Similarly, it could be useful to clarify the difference with FIREs described in Schmidt et al., 2016 (DOI:10.1016/j.celrep.2016.10.061), and/or discuss the synergies and differences with other decomposition approaches for near-diagonal Hi-C interaction patterns (e.g. Zhan et al., 2017, DOI:10.1101/gr.212803.116, Norton et al., 2018, DOI:10.1038/nmeth.4560).

We agree with the reviewers comments. Thank you for the opportunity to add new sections to the discussion to address these points (Lines 412-426).

12. Small points and suggestions:

-- Fig1a: It would very be useful to provide more information about the normalization method used to produce maps (i.e. SHAMAN with k=100 nearest-neighbors), as it is not widely used, as well as the restriction enzyme used for Hi-C (this can help explain the empty bins). It would also be useful to write out the algorithm name in the text when it is cited for the first time.

We have now significantly extended our description of the SHAMAN method and normalization as it was not sufficient (Lines 115-131 in the Methods). The restriction enzyme used for Hi-C is included in the Methods (Line 19).

-- Fig5a: it is unclear if swapping from a 5' to 3' stripe would be changing class or not, given the wording.

We agree that a 5' or 3' stripe are likely fundamentally the same type of TAD structure. We have now included a comment about this in the text (Lines 319-322).

-- FigS5 appears to have a missing panel or an issue with labeling, and a reference appears to be unformatted in the second results section; depositing the manuscript to a preprint server could allow such details to be corrected more quickly.

These are now corrected. Thank you for noticing.

-- The authors write that “the majority of TAD borders were similar (69.4%, separated by less than 50 kb) between pluripotent and lineage committed cells”. It would be interesting to comment more on this in the discussion: is 50kb an experimental limitation (e.g. derived from shifts in boundaries ascertained in different replicates) or do some of these represent biologically important shifts in boundary positioning?

We agree with the reviewer that the question of how to consider TAD border overlap between cell types is a biologically important question. We have now further elaborated this point in the Methods (Lines 221-227). Please note, that our choice of a 50kb overlap was partially to maximise the number of overlapping TAD borders which could then be subsequently classified in domainClassifyR and compared.

-- the use of ‘contact’ to interchangeably describe raw contacts from sequencing as well as high-scoring contact pairs after the SHAMAN procedure leads to some confusing moments in the text. Introducing a term for the latter may improve clarity.

We have now changed the later to ‘high-scoring interactions’ in the text (throughout the manuscript).

-- the code for running domainClassify does not appear to be referred to in the main text or methods. Also, the description of the method is fairly dense, it could help if certain methods or scripts were pointed to in the methods section at the relevant points.

We have now included the link to the GitHub page for domainClassifyR in the methods section (Line 166).

REVIEWERS' COMMENTS:

Reviewer #1 (Remarks to the Author):

I thank the authors for a thorough review and have no additional comments/requests

Reviewer #2 (Remarks to the Author):

The methods in Barrington et al. are greatly clarified. The additional analysis included in the rebuttal would benefit from sharpening. Still, I would be happy to recommend the manuscript for publication.

A number of points remain, which I leave to the discretion of the authors to address:

- in S2L the SHAMAN interaction scores do not appear to have zero mean as a function of increasing genomic distance. In particular, they appear biased at larger genomic distances towards larger values. The authors may want to validate that this does not present an issue for their analysis.
- regarding 2.3 and 2.4: because of the great variability in definitions of TADs (Forcato et al.), it would be useful to list the filtering of TAD sizes to (100kb, 4Mb) in the main text, and that the minimum TAD size for the classification was 200kb (as this is central to the method) in the main text as well, and not just in the methods.
- it appears that the new S4F does not address point 1.5 regarding what happens to individual domains. It may be more informative to keep the same domain classifications from the WT data and plot the mutant interaction scores, rather than try to classify the very small number of domains remaining in the mutant.
- similarly S2D does not really address point 1.1. It would be more informative to consider switching between classifications as in Figure 5A for the additional datasets.
- the authors may want to briefly mention the related and recently-published Kraft et al, 2019 in their discussion doi.org/10.1038/s41556-019-0273-x.
- regarding point 2.10, I appreciate the authors response and believe some of this may benefit the main text discussion as well.

Reviewer #2:

The methods in Barrington et al. are greatly clarified. The additional analysis included in the rebuttal would benefit from sharpening. Still, I would be happy to recommend the manuscript for publication. A number of points remain, which I leave to the discretion of the authors to address:

1. - In S2L the SHAMAN interaction scores do not appear to have zero mean as a function of increasing genomic distance. In particular, they appear biased at larger genomic distances towards larger values. The authors may want to validate that this does not present an issue for their analysis.

Thank you for this question. We believe the reviewer is referring to the plot in S11 where we look at distribution of scores as a function of distance. Apologies if this was unclear. This plot was generated for all intra-TAD data, not for the data specifically used in TAD classification. We have now repeated this for classified TADs (new Supplementary Figure 2a,i). This shows that interaction score means are similar across all distance scales in stripe TADs. Together with the data in panel a) of the same Figure where we confirm that stripe TADs can be identified in the shortest classified TADs, we do not believe this to present an issue with our analysis.

2. - Regarding 2.3 and 2.4: because of the great variability in definitions of TADs (Forcato et al.), it would be useful to list the filtering of TAD sizes to (100kb, 4Mb) in the main text, and that the minimum TAD size for the classification was 200kb (as this is central to the method) in the main text as well, and not just in the methods.

We have now added this to the main text (Line 136) as per the reviewers request.

3. - It appears that the new S4F does not address point 1.5 regarding what happens to individual domains. It may be more informative to keep the same domain classifications from the WT data and plot the mutant interaction scores, rather than try to classify the very small number of domains remaining in the mutant.

We have now adjusted the analysis in S4F to show the interaction scores from the CTCF-aid mutant Hi-C datasets in control classified TAD groups as requested by the reviewer. Indeed, here we can observe differences in TAD architecture in the absence of CTCF, including stripes – further supporting our manuscript and the important role for CTCF in stripe architecture.

4. - Similarly S2D does not really address point 1.1. It would be more informative to consider switching between classifications as in Figure 5A for the additional datasets.

The results currently in S2d show that we can reliably classify TADs from these datasets with similar characteristics as our ESC/NSC data, supporting the computational approach. In addition, we have now repeated the analysis from Figure 5a for the additional Bonev datasets to look at switching between classifications (Response Figure 1). Here, we analysed the serum-grown ESC and NPC Hi-C datasets. We detect a similar 'maintenance' of loop domains between ESC and NPCs and between loop and stripe domains. This result is very interesting because serum-grown ESCs represent a heterogenous population of states of pluripotency, and include cells with a higher probability to exit pluripotency than in homogenous 2i-grown naïve ESCs (the cell type used in our study). Several analyses in the literature show that 2i and serum-grown ESCs have different chromatin, with 2i ESCs exhibiting a 'ground-state' chromatin plasticity. We believe this underlies the differences in our results and further supports the argument that progression through differentiation is associated with topological change and potential.

Response Figure 1. TAD class dynamics between serum-grown ESC (replicate 'A') and NPC (replicate 'A') from Bonev Hi-C datasets. Analysis was performed as for Figure 5a. ESC TADs separated into loop or stripe domains that maintain at least one border during differentiation, and their classification in NPCs. Frequency represents the proportion of NPC classified domains that were maintained or changed from a given ESC TAD group.

5. - the authors may want to briefly mention the related and recently-published Kraft et al, 2019 in their discussion doi.org/10.1038/s41556-019-0273-x.

We have now incorporated this reference into our manuscript.

6. - regarding point 2.10, I appreciate the authors response and believe some of this may benefit the main text discussion as well.

My apologies. We are not sure what point the reviewer is referring to here as we cannot find a point 2.10 in the original review, nor in the first response document.